# Certifying Robust Graph Classification under Orthogonal Gromov-Wasserstein Threats

**Hongwei Jin**[*]
Mathematics and Computer Science Division
Argonne National Laboratory
Lemont, IL 60439
jinh@anl.gov

**Zishun Yu,   Xinhua Zhang**
Department of Computer Science
University of Illinois Chicago
Chicago, IL 60607
{zyu32,zhangx}@uic.edu

## Abstract

Graph classifiers are vulnerable to topological attacks. Although certificates of robustness have been recently developed, their threat model only counts local and global edge perturbations, which effectively ignores important graph structures such as isomorphism. To address this issue, we propose measuring the perturbation with the orthogonal Gromov-Wasserstein discrepancy, and building its Fenchel biconjugate to facilitate convex optimization. Our key insight is drawn from the matching loss whose root connects two variables via a monotone operator, and it yields a tight outer convex approximation for resistance distance on graph nodes. When applied to graph classification by graph convolutional networks, both our certificate and attack algorithm are demonstrated effective.

## 1   Introduction

State-of-the-art graph classifiers such as graph convolutional networks (GCNs) have been recently revealed vulnerable to adversarial structural attacks, namely adding and removing edges [1, 2]. Due to the hardness of finding the strongest attack, an alternative strategy aims to certify robustness, *i.e.*, proving that no perturbation in the threat model can successfully attack the classifier. To this end, [3, 4] extended randomized smoothing to discrete noises with applications in, *e.g.*, community detection. Although our focus is on graph classification, certificates have been studied for node classification using, *e.g.*, convex outer adversarial polytope [5, 6]. Most relevant to our method is [7], which constructed the tightest lower bound of the margin via Fenchel biconjugation.

However, all existing threat models only count the total edges removed or added (global budget) or the change of each node's degree (local budget), while overlooking the important notion of *isomorphism*. In contrast, there has been a wealth of more refined discrepancy measures between graphs, including kernels [8, 9] and GCNs [10, 11]. Recently, the Gromov-Wasserstein discrepancy [GW, 12], which extends the Gromov-Wasserstein distance [13], has emerged as an effective transportation distance between structured data, alleviating the incomparability issue between different structures by aligning the *intra*-relational geometries. Thanks to its favorable properties such as efficiency and awareness of isomorphism, GW has been extensively applied to domain adaptation [14], word embedding [15], graph classification [16], metric alignment [17], generative modeling [18], and graph matching and node embedding [19, 20]. Therefore, it is clearly natural to adopt such a measure in the certification of graph robustness. Analogously, Wasserstein distance has already been employed in distributional robustness [21] and in attacking images [22], where perturbations can be found that better reflect the image content compared with the standard $\ell_p$ attacks.

Unfortunately, GW is not tractable to evaluate, differing from the standard Wasserstein distance which is a linear program. So all existing optimization techniques settle for local solutions, lacking

---

[*]Work done when the author was at the University of Illinois Chicago

an analyzable guarantee. To develop certificates, tractable lower bounds of GW would be essential, but the Kantorovich dual no longer exists and the existing lower bounds [13, 23] are not tight [24]. Further, underlying the standard probability coupling is the metrics within each graph, *e.g.*, shortest-path distance [13, SP distance], which requires nontrivial discrete optimization. This significantly complicates the entire certificate as the subject of perturbation is exactly the topological structure. In addition, certificates based on randomized smoothing [4, 25], which adds noise to the input, are also hindered by the significant challenge in designing the noise distribution that aligns with GW.

The goal of this work, therefore, is to construct the first robustness certificate for GW-style threat models, with applications in graph classification. This is achieved by casting the entire computation procedure of GW as a two-layer model, each of which is subsequently relaxed in a tractable fashion:

$$\text{graph structure} \xrightarrow[\text{matching loss}]{\text{resistance distance}} \text{distance btw nodes} \xrightarrow[\text{convex envelop}]{\text{orthogonal GW}} \text{discrepancy btw graphs.}$$

Firstly, we circumvent the discrete optimization in SP distance by resorting to the resistance distance [26], which, along with its scaled version known as commute time, is one of the most commonly used distances in computer science, machine learning, and beyond [27]. Compared with SP distance, it admits a *closed-form* based on matrix inversion, which is a *monotone* operator on the positive semi-definite cone. This enables our first contribution, where a convex relaxation is designed for resistance distance in Section 3 based on the matching loss [28].

Our second contribution achieves convex relaxation of the GW-style discrepancy measure. Our seed inspiration is drawn from the latest orthogonal GW discrepancy (OGW) [24] reviewed in Section 2.2—it preserves most of the desirable properties of GW while admitting a tight and efficient lower bound. We next make the key observation in Section 2.3 that the lower bound, despite not convex, admits a *closed-form* Fenchel dual, which in turn facilitates an efficient evaluation of the Fenchel biconjugate. Such a convex lower bound is applied to graph convolution networks in Section 4, where both attacking algorithm and robustness certificates are developed.

Our experiments in Section 5 verify the effectiveness of our attacker and certificate, in that a large proportion of the graphs can be proved either vulnerable or robust.

**Related Work**  Certification or verification has been a longstanding pursuit in machine learning and cyber-physical systems. Formally, it inquires if $f(x)$ can be driven down to negative over $x \in \mathcal{X}$, where both $f$ and $\mathcal{X}$ can be too complicated for global optimization. So a lower bound of the optimal objective is sought, which provides incomplete verification (some true properties are confirmed true). A lower bound can also be useful in branch-and-bound algorithms [6]. A natural approach is the Lagrange dual [29, 30], which minorizes the primal by the weak duality. However, it can be loose, and a provably tighter lower bound is the convex envelope or Fenchel biconjugate [7]. But they themselves are efficiently computable only under limited scenarios.

In general, the objective $f$ is composed of several "layers", *i.e.*, $f = f_0 \circ f_1 \circ \ldots$, where $f_i$ is simpler but no longer scalar-valued ($i \geq 1$). So it is natural to relax their graphs. For example, $\min f_0(f_1(x))$ is equivalent to $\min_{x,y} f_0(y)$ s.t. $y = f_1(x)$. So we can possibly use the convex envelop of $f_0$ (which is simpler than $f$) and employ the convex hull of $\{(x, y) : y = f_1(x)\}$. The latter is represented by the convex envelope relaxation of the ReLU activation [31–35], which has also found applications in graph neural networks [6, 36]. However, they have been so far limited to $\mathbb{R} \to \mathbb{R}$ nonlinear functions while a multivariate extension is not available yet, as the convex hull is much harder to compute.

Another commonly used approach is randomized smoothing, which adds noise to the input [25, 37–39]. Although it has also been shown effective in graph robustness [3, 4], the design of discrete noise can be quite intricate and so far no such methods exist for GW-style threat models. Other methods include semi-definite relaxation [40, 41] which is often loose, Lipschitz continuity analysis [42–44] where the local and global estimation of curvature is difficult in the discrete domain, and reformulation linearization technique [45] used by [5] for directed but not undirected graphs.

## 2 Gromov-Wasserstein Threats for Attacking Graph Classifiers

We consider attacking a graph classifier where the edge connectivity is perturbed. Here both the original graph $\mathcal{D}$ and the resulting graph $\mathcal{C}$ are assumed **undirected**, **unweighted**, and **connected**. Let the adjacency matrix of $\mathcal{D}$ be $A \in \{0, 1\}^{n \times n}$, where $A_{ii} = 0$, and $A_{ij} \in \{0, 1\}$ ($i \neq j$) indicating whether an edge exists between nodes $i$ and $j$. Here $n$ is the *order* of the graph, *i.e.*, number of nodes.

Now suppose the graph's edge connectivity is perturbed, leading to a new adjacency matrix $\tilde{A} :=$ $A + X$ (corresponding to $\mathcal{C}$). Obviously, $X$ needs to be symmetric, zero diagonal, and $X_{ij} \in$ $\{-A_{ij}, 1 - A_{ij}\}$. If we relax it to the continuous domain such as convex relaxation, $X_{ij}$ is naturally relaxed to $[-A_{ij}, 1 - A_{ij}]$. The **threat model** governs additional budgets on $X$ besides the above admissible conditions. Global change budget can be written as $\mathbf{1}^\top (X \circ S) \mathbf{1} \leq \delta_g$, where $S = 1 - 2A$ and $\circ$ stands for the Hadamard product. Local budget is $(X \circ S) \mathbf{1} \leq \delta_l \mathbf{1}$ (elementwise). If we only allow adding edges, then $X_{ij} \geq 0$. All these basic constraints are linear in $X$, and we will collectively refer to them as $X \in \mathcal{X}$, a convex set.

Suppose we have trained a graph classifier $\mathcal{G}$ (*e.g.*, graph convolution network, GCN) which maps an adjacency matrix $A$ to a $K$-dimensional logit vector $\mathcal{G}(A)$ corresponding to $K$ classes. Assume the true class is $y$ and $\mathcal{G}$ predicts correctly, *i.e.*, $y = \arg\max_c \mathcal{G}_c(A)$. Then the margin of classification is

$$M(A) := \min_{c \neq y} \{\mathcal{G}_y(A) - \mathcal{G}_c(A)\}. \tag{1}$$

The attacker seeks a feasible perturbation $X \in \mathcal{X}$ that drives the classification margin to negative, *i.e.*, a misclassification. $\mathcal{G}$ is called robust on this graph if $\min_{X \in \mathcal{X}} M(A + X)$ is positive, and is called vulnerable if the minimal value is negative. It is noteworthy that we employ graph classification and GCN only as an example context of applying our proposed relaxation technique. Other tasks such as node classification are readily applicable too – simply replace the $M$ function.

Although the global and local budgets are natural and reasonable characterizations of the perturbation, they ignore an important notion on graphs: isomorphism. Exact isomorphism can be relaxed by defining a distance between two graphs, which quantify their difference under the most favorable permutation of nodes. One of the most commonly used distance is the Gromov-Wasserstein distance [13], which has been extended to Gromov-Wasserstein discrepancy [GW, 12]. So in addition to the standard local and global budgets specified by $\mathcal{X}$, it is natural to further constrain the perturbation in terms of the GW distance. To conclude, the attack and certification tasks can be formalized as $\min_{X \in \mathcal{X}} M(A + X)$ s.t. GW(new graph $A + X$, old graph $A$) $\leq \delta_\Omega$ for some budget $\delta_\Omega > 0$.

## 2.1 Definition of Gromov-Wasserstein distance

Since our threat model will only add or remove edges while the nodes remain intact, we only need to compare two graphs $\mathcal{C}$ and $\mathcal{D}$ with the same order. Although GW takes a prior distribution on nodes that represents their importance, it is often observed that a simple uniform distribution performs equally well or better [12, 16, 46], and many applications do not have such an external prior. So we can restrict GW to uniform distributions, leading to the following expression under $\ell_2$ distance:

$$\mathsf{GW}(\mathcal{C}, \mathcal{D}) = \min_{P \in \mathbb{R}_+^{n \times n} : P\mathbf{1} = P^\top \mathbf{1} = n^{-1}\mathbf{1}} \sum_{i,j,p,q} (C_{ij} - D_{pq})^2 P_{ip} P_{jq}. \tag{2}$$

Here $\mathbf{1}$ is an all-one vector, and $C_{ij}$ is a distance measure between nodes $i$ and $j$ in $\mathcal{C}$, *e.g.*, SP distance. We will denote the $n \times n$ matrices as $C$ (for $\mathcal{C}$) and $D$ (for $\mathcal{D}$). We also follow the idea of [12], where $C_{ij}$ does not have to be a metric and $\ell_2$ can be extended to asymmetric losses. They call it GW discrepancy, and we do not take the square root of the right-hand side of (2) when defining GW. In the sequel, we will also write GW$(C, D)$ instead of GW$(\mathcal{C}, \mathcal{D})$ whenever it causes no confusion.

## 2.2 Orthogonal GW and its upper and lower bounds

Unfortunately, GW relies on a nonconvex optimization, and the optimal $P$ in (2) is intractable to find. Although polytime lower bounds have been proposed by [13], they are shown quite loose [24]. Since our aim is to certify robustness under such a measure, a tight and efficient lower bound is in demand (see below). To this end, we resort to the recently proposed orthogonal GW [OGW, 24], which does offer such a lower bound. OGW first rewrites the GW in the Koopmans-Beckmann form [47]:

$$\mathsf{GW}(C, D) = \tfrac{1}{n^2} \Big( \|C\|_F^2 + \|D\|_F^2 - 2 \max_{P \in \mathcal{E} \cap \mathcal{N}} \mathrm{tr}(CPDP^\top) \Big), \tag{3}$$

$$\text{where} \quad \mathcal{E} := \{P \in \mathbb{R}^{n \times n} : P\mathbf{1} = P^\top \mathbf{1} = \mathbf{1}\}, \quad \mathcal{N} := \mathbb{R}_+^{n \times n}, \quad \|C\|_F^2 := \sum_{ij} C_{ij}^2. \tag{4}$$

Let $\Pi$ be the set of $n \times n$ permutation matrices. The Birkhoff-von Neumann theorem asserts that the domain of coupling ($\mathcal{E} \cap \mathcal{N}$) is the convex hull of $\Pi$. In addition, $\Pi$ can be characterized by

$$\Pi = \mathcal{E} \cap \mathcal{N} \cap \mathcal{O}_n, \quad \text{where} \quad \mathcal{O}_n := \{P \in \mathbb{R}^{n \times n} : P^\top P = PP^\top = I\}. \tag{5}$$

Since both $\mathcal{O}_n$ and $\mathcal{E}$ are spectral constraints, [24] proposed using these two domains only:

$$\mathsf{OGW}(C, D) := \tfrac{1}{n^2}\Big( \|C\|_F^2 + \|D\|_F^2 - 2 \max_{P \in \mathcal{E} \cap \mathcal{O}_n} \operatorname{tr}(CPDP^\top) \Big). \tag{6}$$

To summarize, the attacker tries to minimize the margin by solving the discrete problem:

$$\min_X M(A + X), \quad s.t., \quad X \in \mathcal{X}, \ X_{ij} \in \{-A_{ij}, 1 - A_{ij}\}, \ \mathsf{OGW}(C_{A+X}, C_A) \leq \delta_\Omega, \tag{7}$$

where $C_{A+X}$ and $C_A = D$ are the base distance matrices for $A + X$ and $A$, respectively. Since $\mathsf{OGW}$ is still intractable to compute, we will next review how [24] proposed to approximate it.

**Lower bound of $\mathsf{OGW}$ for certificates**    To certify the robustness, one only needs to optimize (7) with the discrete constraint relaxed into $X_{ij} \in [-A_{ij}, 1 - A_{ij}]$. If the optimal objective value remains above 0, then robustness is certified. To tackle the intractability of $\mathsf{OGW}$, a conservative approach (no false positive) is to relax the feasible domain by replacing $\mathsf{OGW}$ with an efficient *lower bound*.

Although the optimal $P$ in (2) is still intractable to solve, $\mathsf{OGW}$ admits an efficient lower bound that is inspired by the quadratic assignment literature [48]. To begin with, [48] noted that $\mathcal{O}_n \cap \mathcal{E} = \left\{ \tfrac{1}{n}\mathbf{1}\mathbf{1}^\top + VQV^\top : Q \in \mathcal{O}_{n-1} \right\}$, where $V$ is any $n \times (n-1)$ matrix satisfying $V^\top \mathbf{1} = \mathbf{0}$ and $V^\top V = I$. So denoting $\hat{X} := V^\top X V$ and $s_X := \mathbf{1}^\top X \mathbf{1}$ for **any** compatible matrix $X$, we obtain

$$\max_{P \in \mathcal{O}_n \cap \mathcal{E}} \operatorname{tr}(CPDP^\top) = \tfrac{1}{n^2} s_C s_D + \mathcal{Q}(C, D), \tag{8}$$

$$\text{where} \quad \mathcal{Q}(C, D) := \max_{Q \in \mathcal{O}_{n-1}} \{ \operatorname{tr}(\hat{C} Q \hat{D} Q^\top) + \operatorname{tr}(\hat{E}^\top Q) \}, \qquad E := \tfrac{2}{n} C \mathbf{1}\mathbf{1}^\top D. \tag{9}$$

So a lower bound of $\mathsf{OGW}$ (named $\mathsf{OGW}_{lb}$) can be naturally derived by decoupling the two occurrences of $Q$ in $\mathcal{Q}(C, D)$:

$$\mathsf{OGW}_{lb}(C, D) := \tfrac{1}{n^2}\left( \|C\|_F^2 + \|D\|_F^2 - \tfrac{2}{n^2} s_C s_D - 2\mathcal{Q}_{ub}(C, D) \right) \tag{10}$$

$$\text{where} \quad \mathcal{Q}_{ub}(C, D) := \max_{Q_1 \in \mathcal{O}_{n-1}} \operatorname{tr}(\hat{C} Q_1 \hat{D} Q_1^\top) + \max_{Q_2 \in \mathcal{O}_{n-1}} \operatorname{tr}(\hat{E}^\top Q_2). \tag{11}$$

One can derive that $Q_2$ is optimized at $U_E V_E^\top$ where $U_E \Lambda_E V_E^\top$ is the singular value decomposition (SVD) of $\hat{E}$. This results in $\operatorname{tr}(\hat{E}^\top Q_2) = \|\hat{E}\|_*$, the trace norm, which is the sum of its singular values. An optimal $Q_1$ is $P_1 P_2^\top$, if $\hat{C}$ and $\hat{D}$ respectively admit eigen-decompositions $\hat{C} = P_1 \operatorname{diag}(\lambda_1) P_1^\top$ and $\hat{D} = P_2 \operatorname{diag}(\lambda_2) P_2^\top$ [49, 50]. This yields an optimal value of $\operatorname{tr}(\hat{C} Q_1 \hat{D} Q_1^\top)$ as $\lambda_1^\top \lambda_2$. The overall computation costs $O(n^3)$ and the gap arising from decoupling $Q_1$ and $Q_2$ is small because in general, the matrix $\hat{E}$ is much smaller than $\hat{C}$ and $\hat{D}$ in magnitude, as noted by both [48] and [24].

**Upper bound of $\mathsf{OGW}$ for the attacker**    A conservative estimate of whether the attack can be successful is obtainable by further restraining the feasible domain. This can be served by replacing $\mathsf{OGW}$ with its *upper bound*, which can be trivially achieved by locally optimizing $Q$ in (9). We denote it as $\mathsf{OGW}_{ub}$. *Locally* optimizing over $\mathcal{O}_{n-1}$ (*a.k.a.* Stiefel manifold) has been very well studied [51–53], and we adopt a straightforward approach of projected quasi-Newton, noting that the projection of any matrix $Q$ on $\mathcal{O}_{n-1}$ is simply $U_Q V_Q^\top$, where the SVD of $Q$ is $U_Q \Lambda_Q V_Q^\top$.

As is shown by [24], both $\mathsf{OGW}$ and $\mathsf{OGW}_{lb}$ are nonnegative, symmetric, and evaluates zero when $\mathcal{C}$ and $\mathcal{D}$ are isomorphic. Their square root also satisfies the triangle inequality. Experiments on graph classification and barycenter show that they well capture the structural dissimilarities between graphs. Similarly to $\mathsf{GW}$, we compute $\mathsf{OGW}$ in practice via $\mathsf{OGW}_{ub}$.

## 2.3   Convex lower bound of $\mathsf{OGW}$

Although $\mathsf{OGW}_{lb}$ minorizes $\mathsf{OGW}$, it is still not convex in its input $C$, falling short of the requirement of certification. We next derive a convex lower bound via Fenchel biconjugation. In general, biconjugation can be computationally expensive, but interestingly, our case admits efficient recipes.

Recall that in the context of perturbation, we assume $C$ and $D$ correspond to the new and original graphs, respectively, and our goal is to find a convex lower bound for the part of $\mathsf{OGW}(C, D)$ that

depends on $C$, *i.e.*, $f(C) := \|C\|_F^2 - 2\max_{P\in\mathcal{E}\cap\mathcal{O}_n}\mathrm{tr}(CPDP^\top)$. We first compute its Fenchel dual:

$$f^*(R) := \max_C\left\{\mathrm{tr}(R^\top C) - f(C)\right\} = \max_C\left\{\mathrm{tr}(R^\top C) - \|C\|_F^2 + 2\max_{P\in\mathcal{E}\cap\mathcal{O}_n}\mathrm{tr}(CPDP^\top)\right\} \quad (12)$$

$$= \max_{P\in\mathcal{E}\cap\mathcal{O}_n}\max_C\left(\mathrm{tr}(R^\top C) - \|C\|_F^2 + 2\,\mathrm{tr}(CPDP^\top)\right) = \max_{P\in\mathcal{E}\cap\mathcal{O}_n}\left\|\tfrac{1}{2}R + PDP^\top\right\|_F^2 \quad (13)$$

$$= \tfrac{1}{4}\|R\|_F^2 + \|D\|_F^2 + \max_{P\in\mathcal{E}\cap\mathcal{O}_n}\mathrm{tr}(RPDP^\top) \quad (14)$$

$$\leq \tfrac{1}{4}\|R\|_F^2 + \|D\|_F^2 + \tfrac{1}{n^2}s_R s_D + \mathcal{Q}_{ub}(R,D). \quad (15)$$

As $f \geq f^{**}$ ($= (f^*)^*$, the biconjugation), $\mathsf{OGW}(C,D)$ as a function of $C$ can be lower bounded by

$$\mathsf{OGW}(C,D) \geq \tfrac{1}{n^2}\left(\|D\|_F^2 + f^{**}(C)\right) = \tfrac{1}{n^2}\left(\|D\|_F^2 + \max_R\{\mathrm{tr}(R^\top C) - f^*(R)\}\right) \quad (16)$$

$$\geq \tfrac{1}{n^2}\max_R\left\{\mathrm{tr}(R^\top C) - \tfrac{1}{4}\|R\|_F^2 - \tfrac{1}{n^2}s_R s_D - \mathcal{Q}_{ub}(R,D))\right\} \quad (17)$$

$$=: \boxed{\Omega(C)}. \quad (18)$$

The evaluation of $\Omega(C)$ requires solving a convex optimization, utilizing the closed form for $\mathcal{Q}_{ub}$ in (11). In the optimization for the certificate below, we will avoid such an optimization over $R$ by considering the dual of $\Omega(C)$, which can be easily read off from the objective of $R$ in (17).

## 3 Convex Outer Approximation for Resistance Distance

So far, we have taken the distance matrix $C$ as the source of variation. In the certification problem, however, variations originate from the perturbation $X$ on the graph structure. So we will next relate $X$ to $C_{A+X}$, and formulate a *convex* domain in $X$. Unfortunately, this is difficult, so we resort to the convex outer approximation of the graph of $X \mapsto C_{A+X}$. Such an approach has been commonly used in approximating the ReLU activation [31], and we extend it to a multivariate setting via a new approach of matching loss [28].

### 3.1 Using resistance distance as the base measure

Although the SP distance has enjoyed significant popularity, its computation requires a nontrivial discrete optimization, obstructing a convex relaxation. As such, we propose using resistance distance as the base distance because it is a bona fide metric [26], admits a closed form, and costs $O(n^3)$ to compute all-pair distance (same as SP distance). In Section 5.1 and 5.2, we will show that it not only eases computation, but also performs competitively on machine learning tasks. Let the Laplacian of the perturbed graph be $\tilde{L} := \mathrm{diag}(\tilde{A}\mathbf{1}) - \tilde{A}$. Then the resistance distance between node $i$ and $j$ is

$$C_{ij} = \tilde{L}_{ii}^\dagger + \tilde{L}_{jj}^\dagger - 2\tilde{L}_{ij}^\dagger, \quad (19)$$

where $\tilde{L}^\dagger$ is the pseudo-inverse of $\tilde{L}$. Since we only consider connected undirected graphs, it follows $\tilde{L}^\dagger = (\tilde{L} + \frac{1}{n}\mathbf{1}\mathbf{1}^\top)^{-1} - \frac{1}{n}\mathbf{1}\mathbf{1}^\top$. A closely related distance measure is the commute time, which multiply the resistance distance by the volume of the graph ($\mathbf{1}^\top \tilde{A}\mathbf{1}$). Since the volume is a scalar and relates closely with the global budget, it can be incorporated with ease as discussed in Appendix A.1.

In summary, letting $Z = -(\tilde{L} + \frac{1}{n}\mathbf{1}\mathbf{1}^\top)^{-1}$, $C$ is determined through $X \to \tilde{A} \to \tilde{L} \to Z \to \tilde{L}^\dagger \to C$. As both the first two steps and the last two steps are affine, we will denote them as $\tilde{L}_X$ and $C_Z$ respectively. Only the step of $\tilde{L} \to Z$ is nonlinear, posing the major challenge we will address next.

### 3.2 Convex outer approximation of matrix inversion

Let $\mathcal{S}_+$ and $\mathcal{S}_{++}$ be respectively the positive semi-definite and positive definite matrix cones sized $n$-by-$n$. We consider the set

$$\mathcal{F} := \left\{(\tilde{L}, Z) \in \mathcal{S}_+ \times (-\mathcal{S}_{++}) : Z = -(\tilde{L} + \tfrac{1}{n}\mathbf{1}\mathbf{1}^\top)^{-1}, \tilde{L}\mathbf{1} = \mathbf{0}, \tilde{L}_{ij} \leq \beta_{ij}\,\forall i \neq j\right\}.$$

The conditions on $\tilde{L}$ are naturally motivated from graph Laplacian, and it follows implicitly that $\tilde{L} + \frac{1}{n}\mathbf{1}\mathbf{1}^\top \in \mathcal{S}_{++}$. A trivial choice of $\beta_{ij}$ is 0. If we only allow adding edges, then $\beta_{ij}$ can be $-A_{ij}$.

Given a strictly convex function $F$, the associated matching loss is defined as [28]

$$\ell(Z, \Phi) = F^*(Z) + F(\Phi) - \operatorname{tr}(Z^\top \Phi), \qquad (20)$$

and it is well known that the minimum value of $\ell$ is 0, attained if, and only if, $Z = \nabla F(\Phi)$. Now consider $F(\Phi) = -\log\det(\Phi)$ over $\Phi \in \mathcal{S}_{++}$. It is easy to show that $F$ is strictly convex, $\nabla F(\Phi) = -\Phi^{-1}$, and $F^*(Z) = -n - \log\det(-Z)$ over $Z \in -\mathcal{S}_{++}$. $\nabla F^*(Z) = -Z^{-1}$. Therefore, setting $Z = -(\tilde{L} + \frac{1}{n}\mathbf{1}\mathbf{1}^\top)^{-1}$ is equivalent to enforcing

$$\ell(Z, \tilde{L} + \tfrac{1}{n}\mathbf{1}\mathbf{1}^\top) = F^*(Z) + F(\tilde{L} + \tfrac{1}{n}\mathbf{1}\mathbf{1}^\top) - \operatorname{tr}(Z^\top(\tilde{L} + \tfrac{1}{n}\mathbf{1}\mathbf{1}^\top)) \le 0. \qquad (21)$$

This has significantly reduced the difficulty of relaxation because almost all the nonlinearities have been subsumed by the *convex* functions $F^*$ and $F$. The only remaining challenge is the bilinear term in trace, but it is much easier to relax than a general nonlinear function. In particular, by the property of Laplacian, $Z\mathbf{1} = -\mathbf{1}$ and $Z \in -\mathcal{S}_{++}$. Let $T_{ii} = -Z_{ii}$ and $T_{ij} = 2Z_{ij} - Z_{ii} - Z_{jj}$ for $i \ne j$. Then $T \ge \mathbf{0}$ elementwise. So it follows that

$$-\operatorname{tr}(Z^\top \tilde{L}) = -\sum_{i \ne j} \tfrac{1}{2}(T_{ij} - T_{ii} - T_{jj})\tilde{L}_{ij} + \sum_i T_{ii}\tilde{L}_{ii} \qquad (22)$$

$$= -\tfrac{1}{2}\sum_{i \ne j} T_{ij}\tilde{L}_{ij} + \sum_i T_{ii}\sum_j \tilde{L}_{ij} \overset{(\text{by } \tilde{L}\mathbf{1}=\mathbf{0})}{=} -\tfrac{1}{2}\sum_{i \ne j} T_{ij}\tilde{L}_{ij} \ge -\tfrac{1}{2}\sum_{i \ne j} \beta_{ij}T_{ij}. \qquad (23)$$

Let $B$ be a symmetric matrix such that $\operatorname{tr}(BZ) = \frac{1}{2}\sum_{i \ne j}\beta_{ij}T_{ij} = \frac{1}{2}\sum_{i \ne j}\beta_{ij}(2Z_{ij} - Z_{ii} - Z_{jj})$. Then the constraint set $\mathcal{F}$ is enclosed by a *convex* outer approximation as

$$\mathcal{F}_{out} := \{(\tilde{L}, Z) \in \mathcal{S}_+ \times (-\mathcal{S}_{++}) : \ \tilde{L} + \tfrac{1}{n}\mathbf{1}\mathbf{1}^\top \in \mathcal{S}_{++}, \ \tilde{L}\mathbf{1} = \mathbf{0}, \ \tilde{L}_{ij} \le \beta_{ij} \, \forall i \ne j, \qquad (24)$$

$$Z\mathbf{1} = -\mathbf{1}, \ F^*(Z) + F(\tilde{L} + \tfrac{1}{n}\mathbf{1}\mathbf{1}^\top) + 1 - \operatorname{tr}(BZ) \le 0\}. \qquad (25)$$

Overall, we can relax the budget on OGW as follows:

$$\{X : \text{perturbation } X \text{ satisfies } \mathsf{OGW}(C, C_A) \le \delta_\Omega\} \qquad (26)$$

$$= \{X : \mathsf{OGW}(C, C_A) \le \delta_\Omega, \ (\tilde{L}, Z) \in \mathcal{F}, \ C = C_Z, \ \tilde{L} = \tilde{L}_X\} \qquad (27)$$

$$(\text{by } (17)) \ \subseteq \{X : \Omega(C_Z) \le \delta_\Omega, \ (\tilde{L}_X, Z) \in \mathcal{F}\} \qquad (28)$$

$$(\text{by } \mathcal{F} \subseteq \mathcal{F}_{out}) \ \subseteq \{X : \Omega(C_Z) \le \delta_\Omega, \ (\tilde{L}_X, Z) \in \mathcal{F}_{out}\}. \qquad (29)$$

Finally, we attain a convex domain. The last two subsumptions summarize our key contributions of convex relaxation in Section 2.2 and 3, and we will experimentally demonstrate that they are tight.

# 4 Optimization of Attacks and Robustness Certificates with Relaxed OGW

**Attacker algorithm** As discussed in Section 2.2, an attacker solves (7) by resorting to an upper bound of OGW, requiring $\mathsf{OGW}_{ub} \le \delta_\Omega$. Here $\mathsf{OGW}_{ub}$ only needs an efficient local optimization. To deal with the constraint, we resort to the Lagrange dual:

$$\max_{\lambda \ge 0} \min_{X \in \mathcal{X}, X_{ij} \in \{-A_{ij}, 1-A_{ij}\}} M(A + X) + \lambda(\mathsf{OGW}_{ub}(C_{A+X}, C_A) - \delta_\Omega). \qquad (30)$$

Given $\lambda$, $X$ can be optimized greedily as shown in Algorithm 2. To ease notation, we denote $\mathsf{OGW}_{ub}(C_{A+X}, C_A)$ as $\Upsilon(A + X)$. So the problem becomes a simple search for the smallest $\lambda$ such that $\Upsilon(A + X) \le \delta_\Omega$, and it can be solved by binary search as in Algorithm 1. Due to the lack of strong duality, the success of the attack is checked by the sign of $M(A + X)$ at the $X$ found by Algorithm 1, *not* by the sign of the final objective value in (30).

## 4.1 Certificate algorithm

When the GCN has a single hidden layer, [7] showed that $M^*(A)$ can be computed in a closed form over the domain of $A_{ij}$ that is relaxed to $[0, 1]$ $(i \ne j)$, intersected with local and global budgets.

| **Algorithm 1:** Attacking with binary search | **Algorithm 2:** Compute $X_\lambda$ for a given $\lambda$ |
|---|---|
| **Input:** global budget $\delta_g$ and OGW budget $\delta_\Omega$ 
 $\lambda \leftarrow 1$, and compute $X_\lambda$ via Algorithm 2. 
 **if** $\Upsilon(A + X_\lambda) \geq \delta_\Omega$ **then** 
 $\quad$ Double $\lambda$ until $\Upsilon(A + X_\lambda) \leq \delta_\Omega$ 
 $\quad$ $u \leftarrow \lambda$ (upper est.), $l \leftarrow \frac{u}{2}$ (lower est.) 
 **else** 
 $\quad$ Halve $\lambda$ until $\Upsilon(A + X_\lambda) \geq \delta_\Omega$ 
 $\quad$ $l \leftarrow \lambda, u \leftarrow 2l$ 
 **while** $u - l > 0.01$ (or any threshold) **do** 
 $\quad$ $\lambda \leftarrow (u + l)/2$ 
 $\quad$ If $\Upsilon(A + X_\lambda) \geq \delta_\Omega$ **then** $l \leftarrow \lambda$ **else** $u \leftarrow \lambda$ 
 Return $X_\lambda$ | $A_0 \leftarrow A$ 
 **for** $t = 0, 1, \ldots, \delta_g - 1$ **do** 
 $\quad$ $C_t \leftarrow$ cost matrix for $A_t$. 
 $\quad$ $Q_t \leftarrow$ locally optimal $Q$ in $\mathcal{Q}(C_t, C_A)$ 
 $\quad$ **for** $e \in$ *all pairs of nodes* **do** 
 $\quad\quad$ **If** flipping edge $e$ on $A_t$ violates any local 
 $\quad\quad$ budget **then** $V_e \leftarrow \infty$ and continue 
 $\quad\quad$ $A' \leftarrow$ adjacency matrix flipping $e$ on $A_t$ 
 $\quad\quad$ Update $C$ for $A'$ 
 $\quad\quad$ $V_e \leftarrow M(A') + \lambda\Upsilon(A')$, where the $Q$ in 
 $\quad\quad\quad$ $\mathcal{Q}(C, C_A)$ is initialized by $Q_t$ 
 $\quad$ $A_{t+1} \leftarrow$ flip on $A_t$ the edge $\arg\min_e V_e$ 
 Return $X_\lambda := A_{\delta_g} - A$ |

As a result, $M^{**}(A)$ can also be computed efficiently through a convex optimization. Since $M^{**}$ minorizes $M$, we can confirm the robustness (*i.e.*, (7) cannot be reduced to negative) if the lowest possible value of $M^{**}$ is nonnegative over an even *larger* domain of $X$ than that in (7). This is a sufficient but not necessary condition, and the tightness of the relaxation can be verified via the percentage of graphs that can neither be certified as robust, nor successfully attacked by Algorithm 1.

Formalizing this idea, the certification algorithm needs to solve the following problem based on (29),

$$\min_{X,Z} M^{**}(A + X), \quad s.t., \quad X \in \mathcal{X}, \Omega(C_Z) \leq \delta_\Omega, (\tilde{L}_X, Z) \in \mathcal{F}_{out}. \tag{31}$$

Plugging in the definition of $\mathcal{F}_{out}$ and noting that the conditions on $\tilde{L}$ are automatically satisfied by a graph Laplacian, we can explicitize the final *convex* optimization problem as

$$\min_{X \in \mathcal{X}, Z} \quad M^{**}(A + X) \tag{32}$$

$$s.t. \quad \Omega(C_Z) \leq \delta_\Omega, \ Z \in \mathcal{Z} := \{Z \in -\mathcal{S}_{++} : \ Z\mathbf{1} = -\mathbf{1}\} \tag{33}$$

$$F^*(Z) + F(\tilde{L}_X + \tfrac{1}{n}\mathbf{1}\mathbf{1}^\top) + 1 - \text{tr}(BZ) \leq 0. \tag{34}$$

**Optimization algorithm** Solving this optimization efficiently requires appropriately leveraging the structure in the problem. Since both $M^{**}$ and $\Omega$ involve an inner optimization, it will be very inefficient if we nest their evaluation inside the overall procedure. Noting that both $M^*$ and $\Omega^*$ have a closed form, we will dualize these terms and introduce two Lagrange multipliers to enforce the inequalities $\Omega(C_Z) \leq \delta_\Omega$ and (34):

$$\min_{X \in \mathcal{X}, Z \in \mathcal{Z}} \max_{\alpha \geq 0, \gamma \geq 0} \max_{U, S} \text{tr}(U^\top(A + X)) - M^*(U) + \alpha\,\text{tr}(S^\top C_Z) - \alpha\Omega^*(S) \tag{35}$$

$$- \alpha\delta_\Omega + \gamma F^*(Z) + \gamma F(\tilde{L}_X + \tfrac{1}{n}\mathbf{1}\mathbf{1}^\top) - \gamma\,\text{tr}(BZ) + \gamma \tag{36}$$

$$\stackrel{\Psi := \alpha S}{\Longleftrightarrow} \min_{X \in \mathcal{X}, Z \in \mathcal{Z}} \max_{\alpha \geq 0, \gamma \geq 0} \max_{U, \Psi} \text{tr}(U^\top(A + X)) - M^*(U) + \text{tr}(\Psi^\top C_Z) - \alpha\Omega^*(\Psi/\alpha) \tag{37}$$

$$- \alpha\delta_\Omega + \gamma F^*(Z) + \gamma F(\tilde{L}_X + \tfrac{1}{n}\mathbf{1}\mathbf{1}^\top) - \gamma\,\text{tr}(BZ) + \gamma. \tag{38}$$

The introduction of $\Psi$ makes the resulting objective concave in all the max variables $(\alpha, \gamma, U, \Psi)$, noting that $\alpha\Omega^*(\Psi/\alpha)$ is the perspective function of $\Omega^*$, hence convex. This is not the case in (35) because $\alpha\Omega^*(S)$ is not jointly convex. Finally, swapping the min and max in (37), we obtain the dual objective. Fixing the max variables, we can 1) minimize over $X$ by projected quasi-Newton because the objective is smooth and strongly convex, and the domain $\mathcal{X}$ is simple (Appendix A.2); 2) minimize over $Z$, which admits a *closed-form* solution. We defer the details to Appendix A.3. Finally, we maximize over $(\alpha, \gamma, U, \Psi)$ by L-BFGS-B [54], terminating once the dual objective gets positive.

*Remark on connected graph.* As we set up above, the graphs under consideration are always connected. This is the case for all the (original) graphs in our datasets. If a perturbation makes a graph disconnected, then both $\Omega$ and OGW will be infinity. Therefore the attacker in Algorithm 1 will automatically avoid dropping an edge $e$ that disconnects a graph, because the corresponding $V_e$ in Algorithm 2 will be infinity. As for the certificate, since $F(\Phi) = -\log\det(\Phi)$, the term $F(\tilde{L}_X + \tfrac{1}{n}\mathbf{1}\mathbf{1}^\top)$ in (34) creates a barrier to ensure $\tilde{L}_X + \tfrac{1}{n}\mathbf{1}\mathbf{1}^\top$ is positive definite, hence connected.

Table 1: Statistics of the datasets

| dataset | # graphs | # class | # features | ave. edge | min edge | max edge | avg. node | min node | max node |
|---------|----------|---------|------------|-----------|----------|----------|-----------|----------|----------|
| MUTAG | 188 | 2 | 7 | 38 | 20 | 66 | 17.5 | 10 | 28 |
| PTC-MR | 344 | 2 | 18 | 25.0 | 2 | 142 | 13.0 | 2 | 64 |
| COX2 | 467 | 2 | 38 | 86.0 | 68 | 118 | 41.0 | 32 | 56 |
| BZR | 405 | 2 | 56 | 74.0 | 26 | 120 | 35.0 | 13 | 57 |

Table 2: Graph classification accuracy

| | Dataset | Graph Kernels | | GW based SVM | | OGW based SVM | |
|---|---------|-----------|-----------|-----------|-----------|-----------|-----------|
| | | GK (k=3) | WL | SP | RD | SP | RD |
| Vec. | BZR | $78.8 \pm 0.5$ | $78.5 \pm 0.6$ | $78.7 \pm 0.4$ | $79.0 \pm 3.0$ | $78.7 \pm 0.4$ | $78.7 \pm 0.4$ |
| Attr. | COX2 | $78.2 \pm 0.4$ | $78.2 \pm 0.4$ | $78.2 \pm 0.4$ | $78.2 \pm 0.4$ | $78.2 \pm 0.4$ | $78.2 \pm 0.4$ |
| Disc. | MUTAG | $66.5 \pm 0.9$ | $78.8 \pm 4.8$ | $67.5 \pm 2.2$ | $72.1 \pm 4.8$ | $76.1 \pm 4.8$ | $79.8 \pm 5.6$ |
| Attr. | PTC-MR | $61.3 \pm 2.8$ | $61.3 \pm 2.8$ | $56.1 \pm 0.6$ | $56.4 \pm 1.3$ | $57.9 \pm 5.4$ | $59.0 \pm 3.0$ |

# 5 Experimental Results

The goal of our experiments lies in two folds: i) verify that resistance distance (RD) well characterizes graph structures and performs better than or as well as SP distance in classification and barycenter problems, when applied under OGW and GW; ii) demonstrate the effectiveness of our attack and certificate algorithms on real datasets, thereby confirming the tightness of our relaxations. The code is available at [55].

**Datasets.** We experimented on four graph datasets whose statistics are given in Table 1 [56]. They contain a collection of molecules where the vertices represent atoms and edges are chemical bonds. The class label represents certain property of the molecules, *e.g.*, mutagenic effect on a specific bacterium (MUTAG) and carcinogenicity of compounds for male rats (PTC-MR). BZR and COX2 consist of ligands for the benzodiazepine receptor and cyclooxygenase-2 inhibitors, respectively.

## 5.1 Effectiveness of resistance distance in classification

To verify the effectiveness of resistance distance, we first followed [16] to study the graph classification accuracy of an SVM, whose kernel $k(\mathcal{C}, \mathcal{D})$ is defined as $\exp\left(-\gamma \cdot \mathsf{OGW}(\mathcal{C}, \mathcal{D})\right)$ and the base measure for OGW is RD or SP distance. We split the dataset into 75% / 25% for training / testing, and tuned the $\gamma$ and regularizer weight in SVM by 5-fold cross validation on the training set.

The accuracy achieved by SVM based on GW and OGW is presented in Table 2. In addition, we also present graph kernels including graphlet sampling [57] and Weisfeiler-Lehman method [9] as the baselines. Clearly, taking resistance distance as the base measure produces similar or better performance than SP distance under both GW and OGW.

## 5.2 Effectiveness of resistance distance in barycenter

Furthermore, we also validated the resistance distance in the barycenter problem. Given a set of structured data represented as graphs $\{\mathcal{D}_i : i = 1, 2, \ldots, S\}$, it finds the Fréchet mean defined as $\arg\min_{\mathcal{C}} \sum_{i=1}^{S} \lambda_i d(\mathcal{C}, \mathcal{D}_i)$, where $d$ can be GW and OGW, and uniform weights $\lambda_i$ are endowed on all the examples as in [12]. We generated 8 cycle-like graphs with structured noises from order 15 to 25 (Figure 1), and constructed the barycenters of 20 nodes with respect to GW and OGW discrepancies under the base measure of SP and RD. Following Section 3 of [24], we took block coordinate update between $C$ and $P_i$ in (6), where the former has a closed form and the latter is locally optimized inside GW and OGW.

Figure 2 shows the results of constructed barycenter with different discrepancies and base measures. Clearly, both SP and RD capture the key structure property (cycle graph) with some additional structural noise under GW and OGW. It is hard to differentiate which structure is better in the context of noised samples.

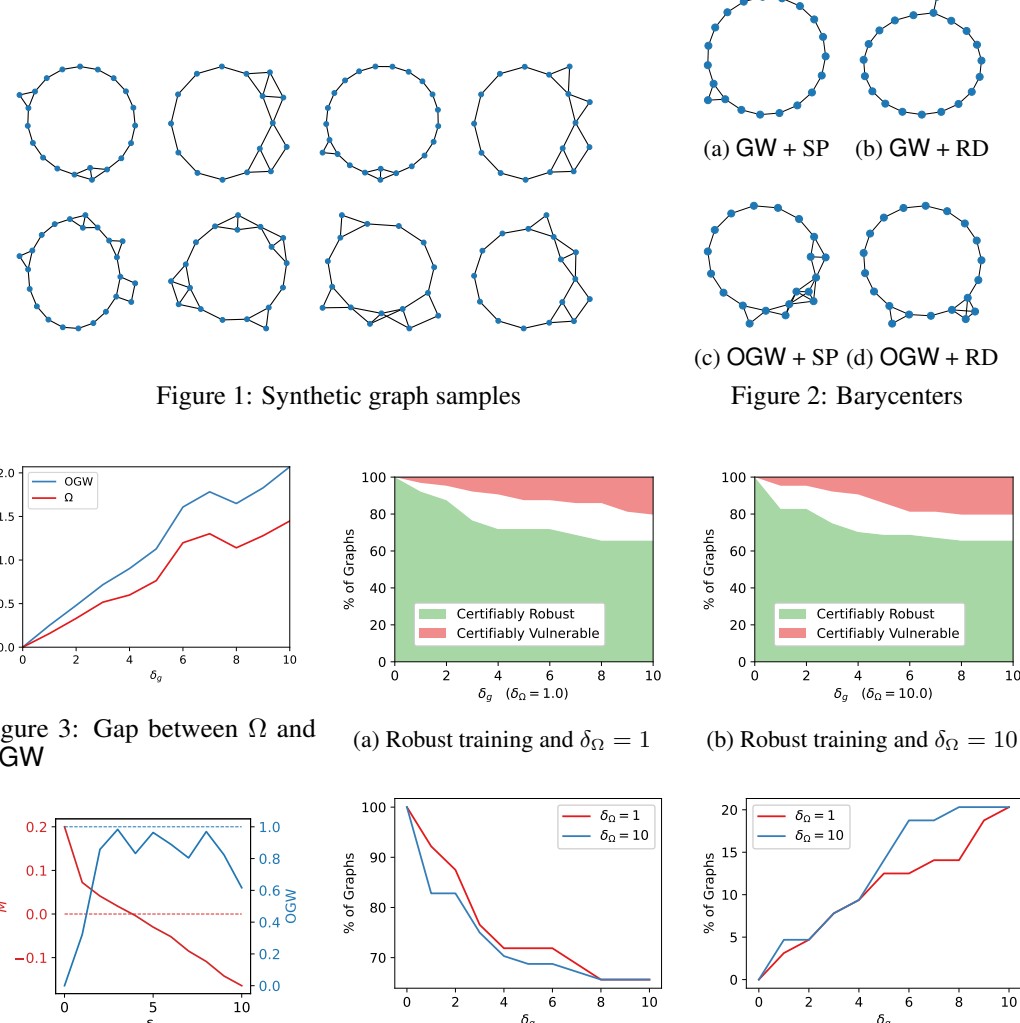

Figure 1: Synthetic graph samples

(a) GW + SP    (b) GW + RD

(c) OGW + SP (d) OGW + RD

Figure 2: Barycenters

Figure 3: Gap between $\Omega$ and OGW

(a) Robust training and $\delta_\Omega = 1$

(b) Robust training and $\delta_\Omega = 10$

Figure 4: $F$ and $\Omega$ in attack on graph No. 125 of MUTAG

(c) Certificate

(d) Attack

Figure 5: Fraction of robust and vulnerable graphs on MUTAG

## 5.3 Performance of certificates and attacks

We next measure the effectiveness of our robustness certificate and attack algorithm jointly, and the criterion is the fraction of graphs that are neither certified as robust nor vulnerable (*i.e.*, successfully attacked). Due to space limitation, we only report the result of MUTAG in the main paper, and defer the result of the other three datasets to Appendix B. To the best of our knowledge, there is no existing certification algorithm that addresses GW or OGW budgets, except generic global optimization techniques which do not scale well when nested inside the overall problem (7).

**Settings.** Following [7], we split each dataset into 30%, 20%, and 50% for training, validation, and testing, respectively. A GCN model was then learned using a single linear convolutional layer with 64 hidden nodes, followed by average pooling. [7] also constructed $M^{**}$ with ReLU activation, but showed larger gap than linear activation. Since our focus here is on the tightness of certificate and attack under OGW, we would like to insulate the complication from ReLU and hence focused on linear activation. Following [5, 36], the GCN is trained with a hinge loss that promotes large margin from (1) for robustness: $\sum_{c \neq y} \max\{0, 1 + \max_A \{\mathcal{G}_c(A) - \mathcal{G}_y(A)\}\}$, where $A$ is optimized under the budget of $\delta_l = 1$ and $\delta_g = 10$.

**Results.** To start with, we empirically checked the tightness of convexified OGW. We took the first graph in the MUTAG dataset, and randomly perturbed its structure (adding or deleting edges) for 20 times with the global budget $\delta_g$ varied from 1 to 10. Figure 3 shows the average values of OGW and $\Omega$ with resistance distance used as the base measure. As $\delta_g$ increases, both OGW and $\Omega$ grow, but not monotonically. This is consistent with the fact that perturbing multiple edges may just lead to small changes in GW and OGW due to isomorphism. Moreover, the gap between OGW and $\Omega$ gets wider with higher $\delta_g$, but their relative difference remains almost intact.

Figure 5 shows the fraction of certifiably robust graphs in the lower green area, and 100 minus the fraction of certified vulnerable graphs in the upper red area. This leaves the white area showing the gap of undetermined graphs, and a narrow white area indicates the effectiveness of both the certificate and the attacker. Here, we fixed $\delta_l = 1$, which is sufficient for the global budget $\delta_g$ on MUTAG.

Sub-plots (a) and (b) present the result for $\delta_\Omega = 1$ and 10, respectively. The gap grows with $\delta_g$ and stays below 20% of the graphs. Sub-plots (c) and (d) provide a clearer comparison, between different values of $\delta_\Omega$, over the performance of certificate and attack. Increasing the value of $\delta_\Omega$ enlarges the feasible domain in (33), allowing more (less) graphs to be certified vulnerable (robust).

Figure 4 shows, for graph No. 125 in MUTAG, the value of OGW and $M$ as $\delta_g$ is increased and $\delta_\Omega = 1$. Interestingly, the increase of $\delta_g$ does not monotonically consume more budget in $\Omega$, which is consistent with the property of $\Omega$.

We additionally compared the performance of our vanilla and robust one-layer GCN model with an MLP model with node feature only, and two other state-of-the-art models, namely MemGNN [58] and FactorGCN [59]. The results are deferred to Appendix E.

## 6 Conclusion

We designed a new convex lower bound for the orthogonal Gromov-Wasserstein distance based on Fenchel biconjugation. Combined with a convex relaxation of the resistance distance using matching loss, it provides a tight certificate of robustness for graph classification with GCNs.

Future work can extend the approach to certifying distributional robustness, and further leverage the closed form of resistance distance for provable optimization on graphs. It is noteworthy that our relaxation of OGW with resistance distance is orthogonal to the graph classifier itself. When the classifier is changed, one will only need to re-derive the expression of $M^{**}$ for the risk or margin. The constraints in (30) and (32), however, remain intact, and that part is our contribution. It will also be interesting to extend GCN to multiple layers GCN, by, *e.g.*, leveraging the convex envelop discussed in Appendix D of [7].

## Acknowledgments and Disclosure of Funding

This material is based upon work supported by the National Science Foundation under Grant No. 1910146.

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
