# Supplementary Material

All code and data are available at

`https://github.com/cshjin/cert_ogw`

## A   Proofs and detailed algorithms

### A.1   Discussion on commute time

The commute time between node $i$ and $j$ is $\pi$ times their resistance distance, where $\pi$ is the volume of the graph defined as $\pi := \mathbf{1}^\top \tilde{A} \mathbf{1}$. We can treat $\pi$ as a constant, because it is equal to $\mathbf{1}^\top A \mathbf{1} + \mathbf{1}^\top X \mathbf{1}$, and we can enumerate the value of $\mathbf{1}^\top X \mathbf{1}$ for optimization over $X$. When we only allow adding edges, this corresponds exactly to the global budget of new edges to add. Overall, the range of $\mathbf{1}^\top X \mathbf{1}$ is narrow because the global budget is low.

### A.2   Projection to $\mathcal{X}$

Let $s_{ij} = 1$ if $H_{ij} = 0$, and $s_{ij} = -1$ if $H_{ij} = 1$. Set $s_{ii} = 0$. It is easy to show that

$$\mathcal{X} = \{X \in \mathbb{R}^{n \times n} : X_{ii} = 0,\ X_{ij} + H_{ij} \in [0, 1],\ \text{tr}(SX) \le 2\delta_g\}. \tag{39}$$

To project an $X^{(t)} \in \mathbb{R}^{n \times n}$ to $\mathcal{X}$, we alternate between two projections:

1. Project to

$$\{X \in \mathbb{R}^{n \times n} : X_{ii} = 0,\ X_{ij} + H_{ij} \in [0, 1]\}. \tag{40}$$

Denote the projection image as $Z$. Then $Z_{ii}$ to 0, and

$$Z_{ij} = \text{median}(X_{ij}^{(t)}, -H_{ij}, 1 - H_{ij}). \tag{41}$$

2. Project $Z$ to

$$\{X \in \mathbb{R}^{n \times n} : X_{ii} = 0,\ \text{tr}(SX) \le 2\delta_g\}. \tag{42}$$

The resulting $X^{\text{proj}}$ is

$$X^{\text{proj}} = \begin{cases} Z & \text{if } \text{tr}(SZ) \le 2\delta_g \\ Z - \|S\|^{-2}\left(\text{tr}(SZ) - 2\delta_g\right)S & \text{otherwise} \end{cases}. \tag{43}$$

We can terminate the alternating between 1 and 2 when $\left\|X^{\text{proj}} - X^{(t)}\right\|$ falls below some threshold. Usually 20 rounds will be sufficient.

### A.3   Closed-form Solution for $Z$ in (37)

We copy (37) for convenience:

$$\max_{\alpha \ge 0, \gamma \ge 0} \max_{U, \Psi} \min_{X \in \mathcal{X}, Z \in \mathcal{Z}} \text{tr}(U^\top(A + X)) - M^*(U) + \text{tr}(\Psi^\top C_Z) - \alpha\Omega^*(\Psi/\alpha) \tag{44}$$

$$- \alpha\delta_\Omega + \gamma F^*(Z) + \gamma F(\tilde{L}_X + \tfrac{1}{n}\mathbf{1}\mathbf{1}^\top) - \gamma\,\text{tr}(BZ) + \gamma \tag{45}$$

Given $(\alpha, \gamma, U, \Psi)$, the optimization over $Z$ is

$$\min_{Z: Z \prec \mathbf{0}, Z\mathbf{1} = -\mathbf{1}} \gamma^{-1}\,\text{tr}(\Psi^\top C_Z) + F^*(Z) - \text{tr}(BZ). \tag{46}$$

It is easy to see that $C_Z$ is linear in $Z$ (not only affine):

$$C_Z = -Z_{ii} - Z_{jj} + 2Z_{ij}. \tag{47}$$

So we can define a linear operator $\mathcal{A}$ as $\mathcal{A}(Z) = C_Z$, and then $\mathrm{tr}(\Psi^\top C_Z) = \mathrm{tr}(\mathcal{A}^*(\Phi)Z)$, where $\mathcal{A}^*$ is a adjoint of $\mathcal{A}$ and can be derived as follows for a symmetric $\Phi$:

$$\mathcal{A}^*(\Psi) = \begin{cases} 2\Psi_{ij} & \text{if } i \neq j \\ -2\sum_j \Psi_{ij} & \text{if } i = j \end{cases}. \tag{48}$$

So clearly $\mathcal{A}^*(\Psi)\mathbf{1} = \mathbf{0}$. By the definition of $B$ in Section 3.2, we also derive $B\mathbf{1} = \mathbf{0}$.

We next use Lagrangian multiplier $\mu$ to enforce $Z\mathbf{1} = -\mathbf{1}$. To ensure symmetry, we equivalently enforce $Z\mathbf{1} + Z^\top \mathbf{1} + 2\mathbf{1} = \mathbf{0}$. Letting $J = B - \gamma^{-1}\mathcal{A}^*(\Psi)$, the Lagrangian of (46) becomes

$$\max_\mu \min_{Z \prec \mathbf{0}} \ \gamma^{-1}\mathrm{tr}(\Psi\mathcal{A}(Z)) + F^*(Z) - \mathrm{tr}(BZ) - \mu^\top(Z\mathbf{1} + Z^\top\mathbf{1} + 2\mathbf{1}) \tag{49}$$

$$= -\min_\mu \max_{Z \prec \mathbf{0}} \left\{ \mathrm{tr}((J + \mu\mathbf{1}^\top + \mathbf{1}\mu^\top)Z) - F^*(Z) + 2\mathbf{1}^\top\mu \right\} \tag{50}$$

$$= -\min_\mu \left\{ F(J + \mu\mathbf{1}^\top + \mathbf{1}\mu^\top) + 2\mathbf{1}^\top\mu \right\}. \tag{51}$$

The optimal $Z$ is

$$J + \mu\mathbf{1}^\top + \mathbf{1}\mu^\top = \nabla F^*(Z) = -Z^{-1} \qquad \Rightarrow \qquad Z = -(J + \mu\mathbf{1}^\top + \mathbf{1}\mu^\top)^{-1}. \tag{52}$$

To solve $\mu$, notice $J\mathbf{1} = \mathbf{0}$. Taking the derivative of (51) with respect to $\mu$, we get

$$-2(J + \mu\mathbf{1}^\top + \mathbf{1}\mu^\top)^{-1}\mathbf{1} + 2\mathbf{1} = \mathbf{0} \qquad \Rightarrow \qquad \mu = \tfrac{1}{2n}\mathbf{1}. \tag{53}$$

This implies that the optimal $Z$ is

$$Z^\star = -(J + \tfrac{1}{n}\mathbf{1}\mathbf{1}^\top)^{-1} = -(B - \gamma^{-1}\mathcal{A}^*(\Psi) + \tfrac{1}{n}\mathbf{1}\mathbf{1}^\top)^{-1}. \tag{54}$$

# B   More Experimental Results

## B.1   Results of certificate and attack from other datasets than MUTAG

As part of Section 5.3, the certificate and attack results for BZR, COX2 and PTC-MR are shown in Figure 6 to 8. They corroborate and reinforce the conclusions in Section 5.3. Due to the variance of characteristics in different datasets, we fixed $\delta_l = 5, 5, 1$ for the datasets BZR, COX2 and PTC-MR, respectively.

## B.2   Computational complexity

We measured the wall-clock time of certificate from the MUTAG dataset. Figure 9a shows the runtime of each iteration for graphs with different orders. As described in Section 4.1 and Appendix A.2 and A.3, the per-iteration cost of the optimization is $O(n^3)$. Figure 9b further shows the total time taken to find a certificate, *i.e.*, a positive value from our dual objective (44), noting that we can early-stop once a positive value is reached. Overall, the cost is mild.

We implemented the algorithm in Python with wrapped L-BFGS-B algorithm from Scipy, and ran the experiments on a machine with Intel CPU i9-9900X.

# C   Reachable graphs given specific budgets

Without the constraints of local and global budget, checking all the reachable graphs essentially finds all possible perturbations under the budget $\delta_\Omega$ on $\Omega$, which is (NP) hard. Alternatively, we examined the $\Omega$ value on the real dataset MUTAG by extracting all the graphs with 12 nodes, and then presented their pairwise $\Omega$ distance (first line in each cell) and $\delta_g$ (second line) in Figure 10.

To better visualize the result, Figure 11 and 12 set the budget $\delta_\Omega$ to 0.5 and 1 respectively, and a darker shade represents a higher value of $\Omega$. A cell is marked with two numbers (red for $\Omega$ and black for #perturbed-edge) computed from a pair of reachable graphs, if its $\Omega$ value falls below the $\delta_\Omega$ budget. In Figure 11, we observe that in the first row, the columns 5, 10, 11, 12, 13 exhibit high values of #perturbed-edge, but their $\Omega$ value is 0. In these cases the pair of graphs are isomorphic, although their topology differs a lot. We also see a block of four isomorphic graphs in the bottom-right corner.

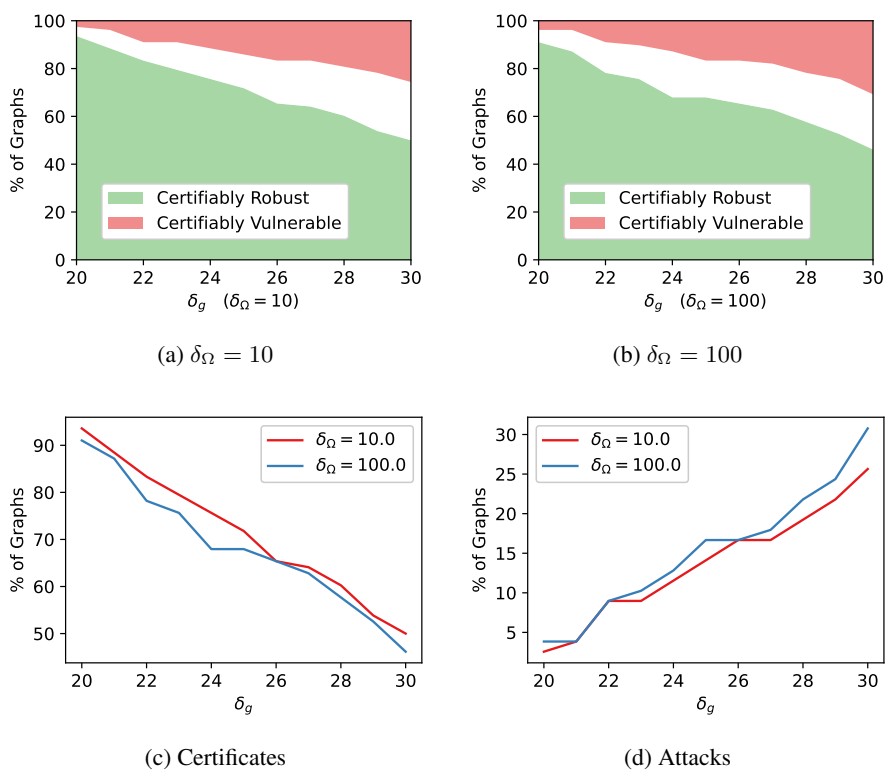

(a) $\delta_\Omega = 10$        (b) $\delta_\Omega = 100$

(c) Certificates        (d) Attacks

Figure 6: Certificate and attack on BZR ($\delta_l = 5$)

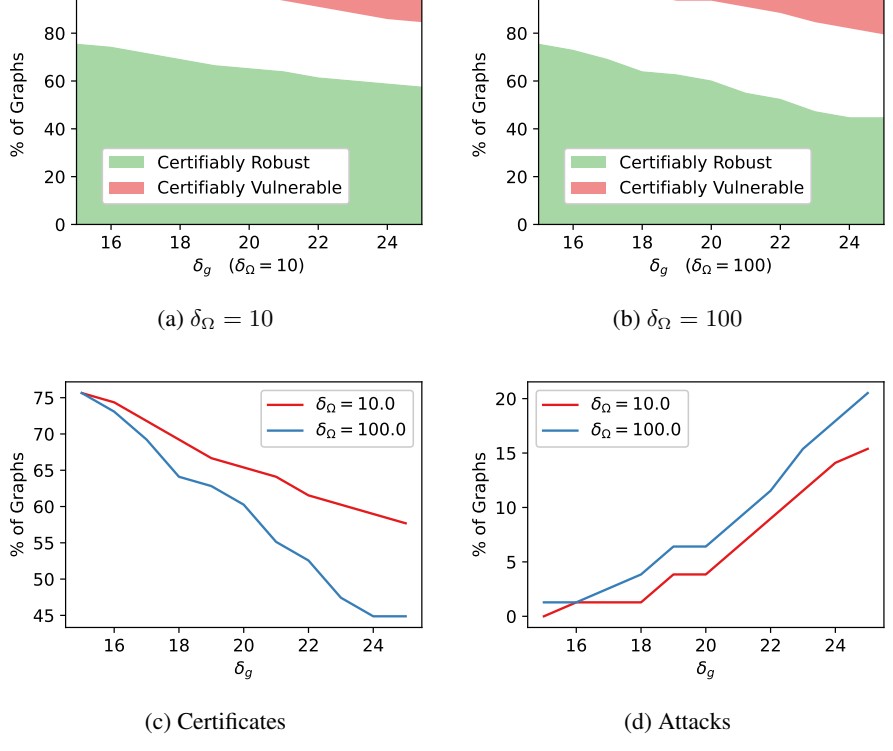

(a) $\delta_\Omega = 10$        (b) $\delta_\Omega = 100$

(c) Certificates        (d) Attacks

Figure 7: Certificate and attack on COX2 ($\delta_l = 5$)

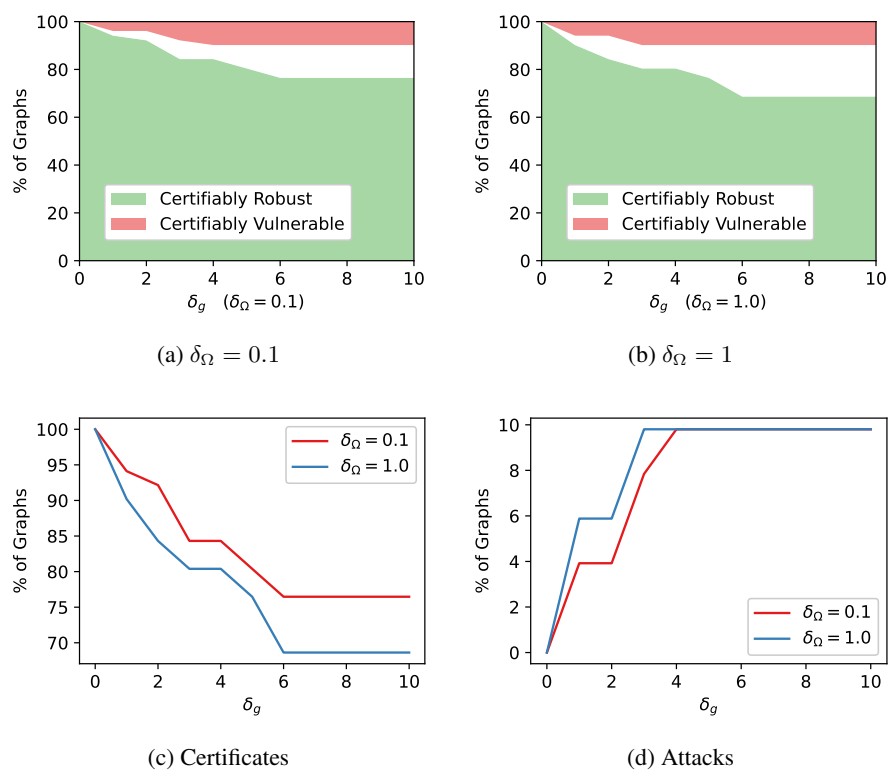

(a) $\delta_\Omega = 0.1$

(b) $\delta_\Omega = 1$

(c) Certificates

(d) Attacks

Figure 8: Certificate and attack on PTC-MR ($\delta_l = 1$)

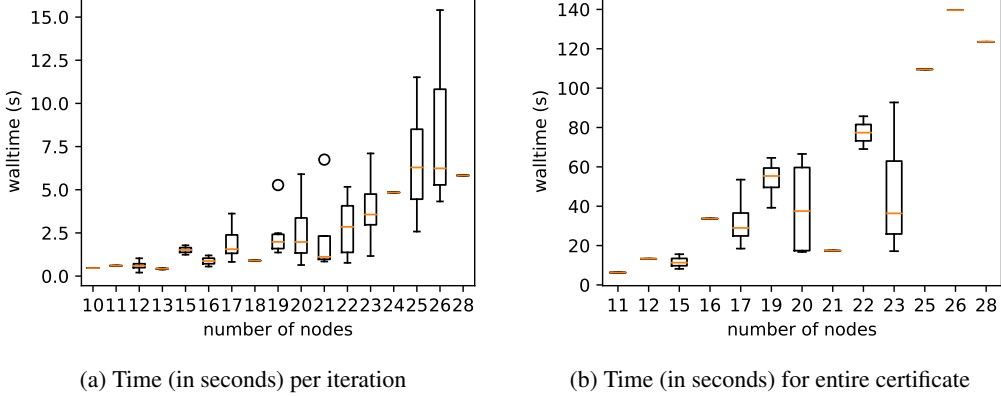

(a) Time (in seconds) per iteration

(b) Time (in seconds) for entire certificate

Figure 9: Running time for certification (MUTAG)

Comparing Figure 12 with Figure 11, clearly more pairs of graphs become reachable thanks to the increase in $\delta_\Omega$.

Similarly, Figure 13 and 14 set the threshold of $\delta_g$ to $4$ and $8$ respectively, and a darker shade represents a higher value of #perturbed-edge. A cell is marked with two numbers (red for #perturbed-edge and black for $\Omega$) computed from a pair of reachable graphs, if its #perturbed-edge falls below the $\delta_g$ budget.

| $\delta_\Omega$ / $\delta_g$ | 1 | 2 | 3 | 4 | 5 | 6 | 7 | 8 | 9 | 10 | 11 | 12 | 13 |
|---|---|---|---|---|---|---|---|---|---|---|---|---|---|
| 1 | 0.00 / 0 | 0.36 / 7 | 0.24 / 9 | 0.36 / 7 | 0.00 / 10 | 2.93 / 3 | 2.01 / 5 | 0.93 / 2 | 0.39 / 10 | 0.00 / 10 | 0.00 / 4 | 0.00 / 10 | 0.00 / 10 |
| 2 | 0.36 / 7 | 0.00 / 0 | 0.37 / 2 | 0.00 / 8 | 0.36 / 5 | 3.82 / 6 | 1.10 / 6 | 3.62 / 7 | 0.42 / 13 | 0.36 / 5 | 3.78 / 5 | 0.36 / 5 | 0.36 / 5 |
| 3 | 0.24 / 9 | 0.37 / 2 | 0.00 / 0 | 0.38 / 10 | 0.24 / 7 | 2.22 / 8 | 2.44 / 8 | 0.36 / 9 | 3.15 / 15 | 0.24 / 7 | 2.99 / 7 | 0.24 / 7 | 0.24 / 7 |
| 4 | 0.36 / 7 | 0.00 / 8 | 0.38 / 10 | 0.00 / 0 | 0.56 / 11 | 2.15 / 4 | 1.68 / 6 | 3.65 / 7 | 0.42 / 13 | 0.56 / 11 | 0.56 / 7 | 0.56 / 11 | 0.56 / 11 |
| 5 | 0.00 / 10 | 0.36 / 5 | 0.24 / 7 | 0.56 / 11 | 0.00 / 0 | 3.32 / 9 | 1.71 / 9 | 0.60 / 10 | 0.14 / 12 | 0.00 / 0 | 0.00 / 10 | 0.00 / 0 | 0.00 / 0 |
| 6 | 2.93 / 3 | 3.82 / 6 | 2.22 / 8 | 2.15 / 4 | 3.32 / 9 | 0.00 / 0 | 1.06 / 4 | 0.39 / 5 | 2.82 / 11 | 2.90 / 9 | 3.07 / 5 | 2.90 / 9 | 2.90 / 9 |
| 7 | 2.01 / 5 | 1.10 / 6 | 2.44 / 8 | 1.68 / 6 | 1.71 / 9 | 1.06 / 4 | 0.00 / 0 | 1.83 / 5 | 2.26 / 11 | 2.85 / 9 | 2.68 / 5 | 2.85 / 9 | 2.85 / 9 |
| 8 | 0.93 / 2 | 3.62 / 7 | 0.36 / 9 | 3.65 / 7 | 0.60 / 10 | 0.39 / 5 | 1.83 / 5 | 0.00 / 0 | 0.51 / 8 | 0.60 / 10 | 0.60 / 4 | 0.60 / 10 | 0.60 / 10 |
| 9 | 0.39 / 10 | 0.42 / 13 | 3.15 / 15 | 0.42 / 13 | 0.14 / 12 | 2.82 / 11 | 2.26 / 11 | 0.51 / 8 | 0.00 / 0 | 0.31 / 12 | 0.31 / 12 | 0.31 / 12 | 0.31 / 12 |
| 10 | 0.00 / 10 | 0.36 / 5 | 0.24 / 7 | 0.56 / 11 | 0.00 / 0 | 2.90 / 9 | 2.85 / 9 | 0.60 / 10 | 0.31 / 12 | 0.00 / 0 | 0.00 / 10 | 0.00 / 0 | 0.00 / 0 |
| 11 | 0.00 / 4 | 3.78 / 5 | 2.99 / 7 | 0.56 / 7 | 0.00 / 10 | 3.07 / 5 | 2.68 / 5 | 0.60 / 4 | 0.31 / 12 | 0.00 / 10 | 0.00 / 0 | 0.00 / 10 | 0.00 / 10 |
| 12 | 0.00 / 10 | 0.36 / 5 | 0.24 / 7 | 0.56 / 11 | 0.00 / 0 | 2.90 / 9 | 2.85 / 9 | 0.60 / 10 | 0.31 / 12 | 0.00 / 0 | 0.00 / 10 | 0.00 / 0 | 0.00 / 0 |
| 13 | 0.00 / 10 | 0.36 / 5 | 0.24 / 7 | 0.56 / 11 | 0.00 / 0 | 2.90 / 9 | 2.85 / 9 | 0.60 / 10 | 0.31 / 12 | 0.00 / 0 | 0.00 / 10 | 0.00 / 0 | 0.00 / 0 |

Figure 10: Pairwise $\Omega$ distance and $\delta_g$

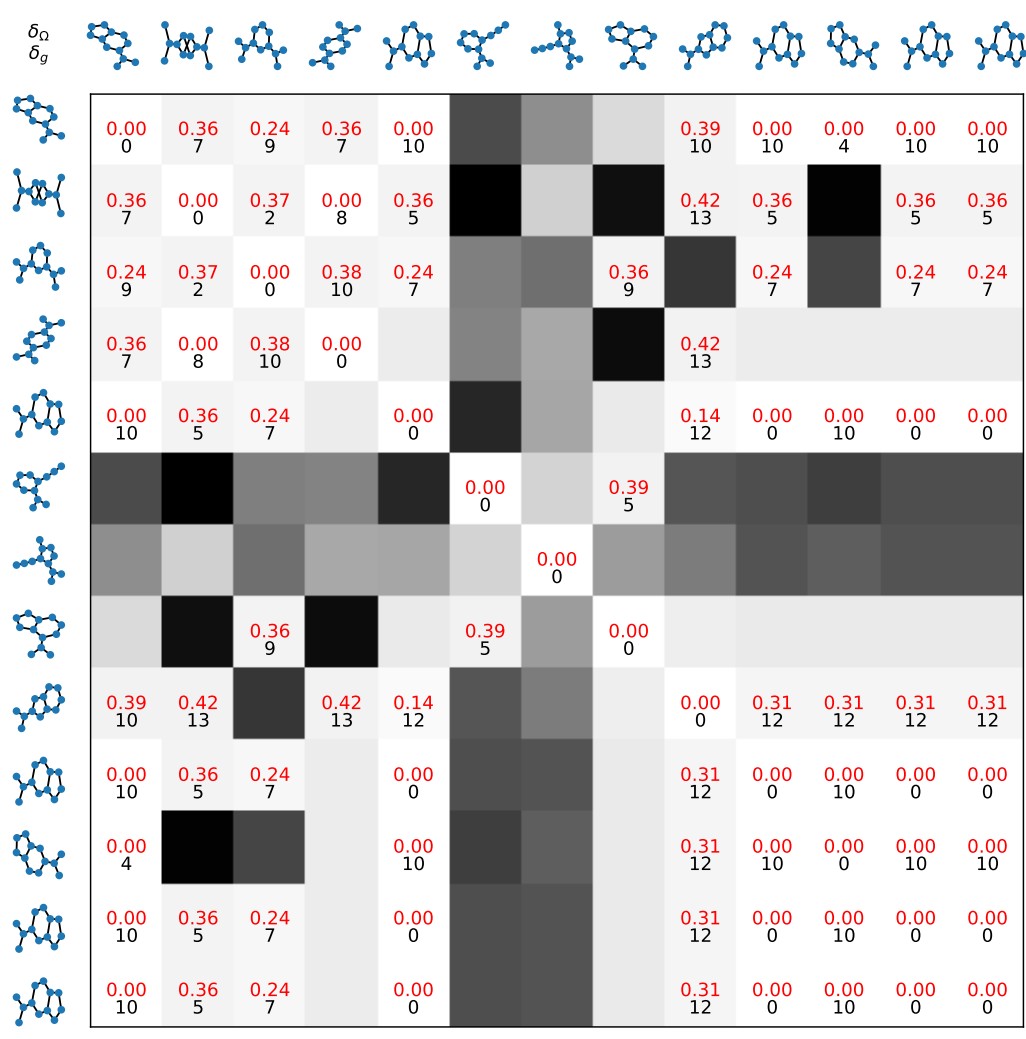

Figure 11: Reachable graphs given $\delta_\Omega = 0.5$

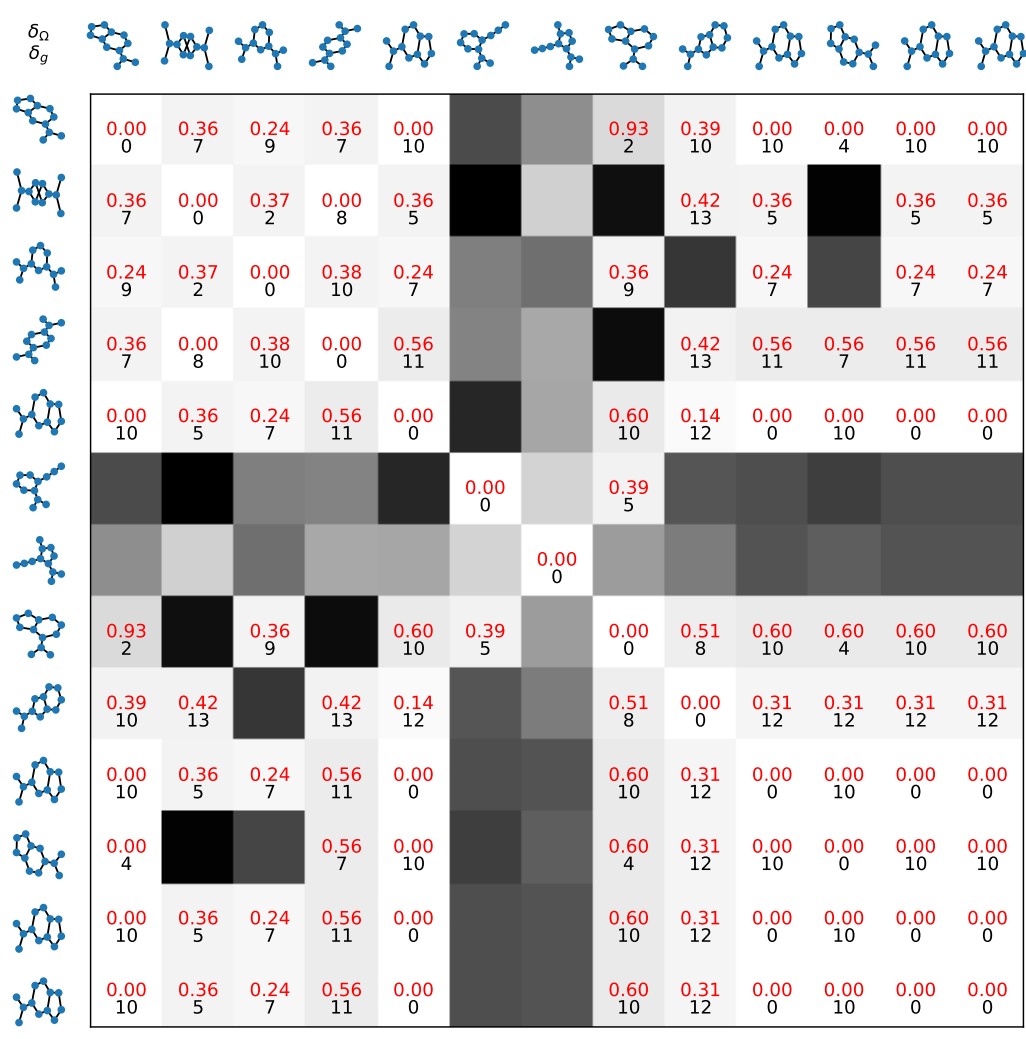

Figure 12: Reachable graphs given $\delta_\Omega = 1$

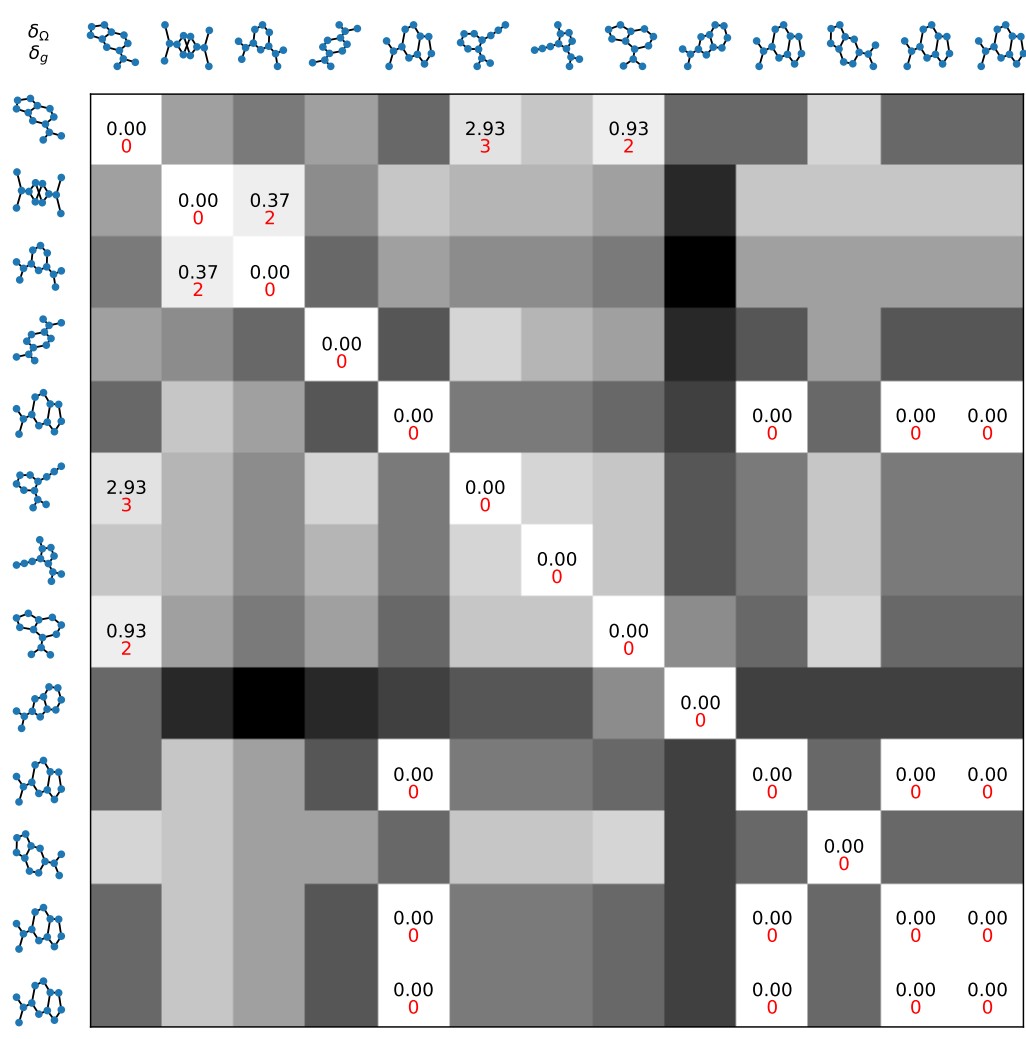

Figure 13: Reachable graphs given $\delta_g = 4$

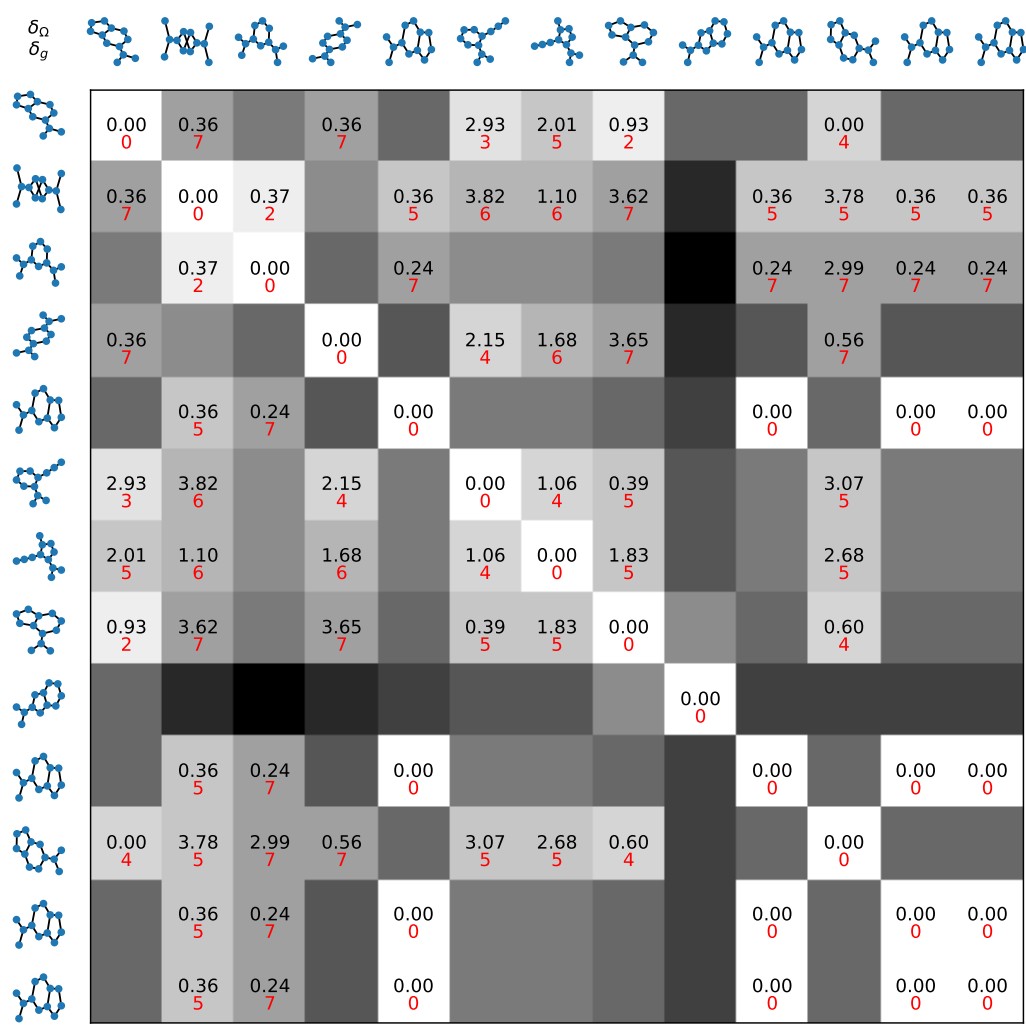

Figure 14: Reachable graphs given $\delta_g = 8$

# D    Plots for Better Turned Hyperparameter

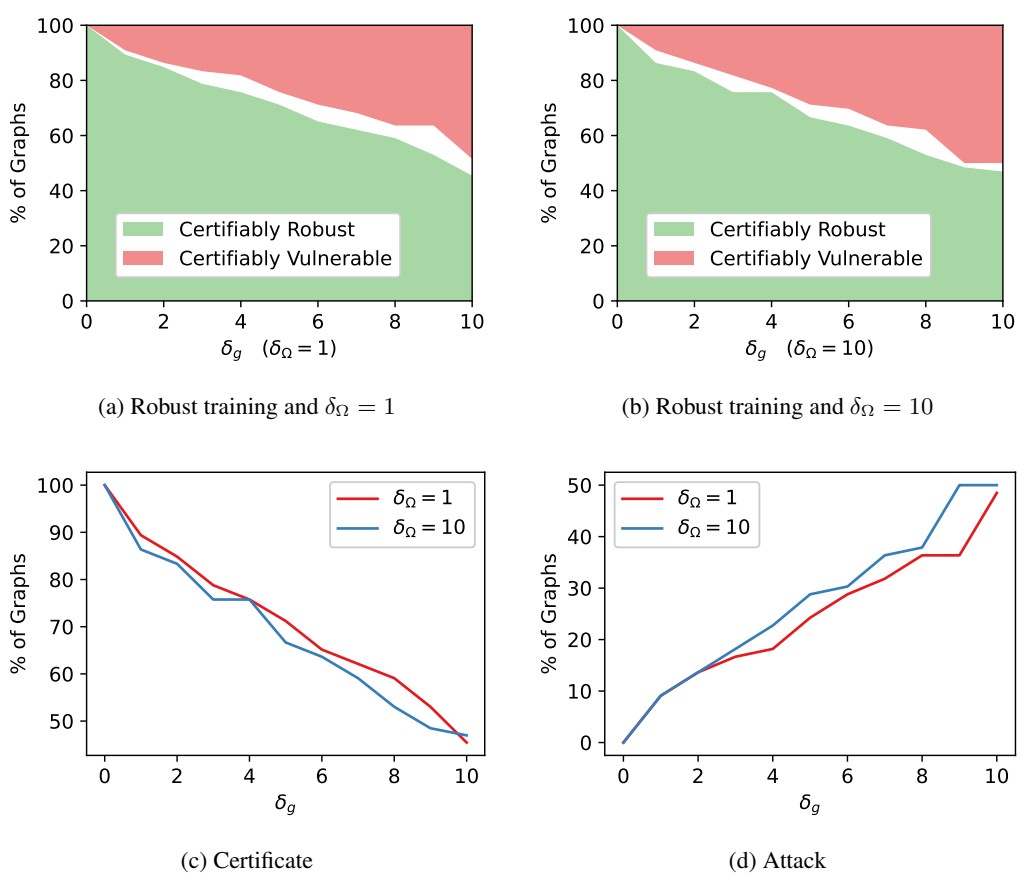

(a) Robust training and $\delta_\Omega = 1$

(b) Robust training and $\delta_\Omega = 10$

(c) Certificate

(d) Attack

Figure 15: Fraction of robust and vulnerable graphs on MUTAG (with tuned hyperparameter)

# E    Comparison of Classification Performance

Table 3: Performance of Classification

| Dataset | Vanilla-GCN | Robust-GCN | MLP | MemGNN | FactorGCN |
|---------|-------------|------------|-----|--------|-----------|
| BZR | 81.8 | 80.3 | 79.9 | 84.7 | 82.4 |
| COX2 | 79.9 | 78.6 | 78.2 | 79.0 | 81.9 |
| MUTAG | 69.5 | 67.4 | 65.0 | 77.8 | 82.6 |
| PTC_MR | 57.8 | 57.8 | 57.3 | 59.8 | 54.6 |

We compared the performance of our vanilla and robust one-layer GCN model with a MLP model with node feature only, and two other models, namely MemGNN [58] and FactorGCN [59]. To be consistent with our setting, we split the training, validation and test sets into 30, 20, and 50% respectively. All the other hyperparameters followed the standard setting from the papers. Table 3 reports the average accuracy on the test set with 5 runs, where most times the robust model sacrifices only a slight amount of accuracy compared with our vanilla model.