# OpenReview forum: "Certifying Robust Graph Classification under Orthogonal Gromov-Wasserstein Threats"
_NeurIPS.cc/2022/Conference — NeurIPS 2022 Accept_

### Official Review · Reviewer_hBdY · 2022-07-10

**Rating:** 6
**Confidence:** 5
**Soundness:** 4 excellent
**Presentation:** 3 good
**Contribution:** 3 good

**Summary:**

The authors propose a robustness certificate for graph classifiers under orthogonal Gromov-Wasserstein threats which are sensitive to whether two graphs are isomorphic. They use a (convex relaxation) of the resistance distance as the underlying metric, and derive a convex lower bound of OGW via Fenchel biconjugation. This gives them a sound but incomplete certificate. They also propose a complementary attack to show non-robustness. They evaluate their certificate on graph classification tasks using a single layer linear GCN.

**Questions:**

1.  Can the certificate be applied to multi-layer GCNs, and if yes, how do the results change?
2.  How do the plots in Fig. 3 change for larger values of local budget?
3.  Given very large (practically infinite) values of local and global budget and some seed original graph, what are all the reachable graphs that satisfy the OGW constraint? It would be helpful to provide a few examples, starting with small values of $\delta_\Omega$ and gradually increasing it.
4. Same as above, but now also enforcing the local and global budgets.
5. What is the accuracy of the certified model in section 5.3. and how does it compare to SOTA (non-certifiable) models?
6. How do vanilla-trained models perform and does robust training decrease the clean accuracy?

**Limitations:**

The authors do not explicitly discuss the limitations of the limitations and potential negative societal impact of their work.

**Strengths And Weaknesses:**

The paper is very well written and easy to follow. The problem that they tackle is very important since it is moving us beyond simple threat models such as global and local edge perturbations that effectively ignore the graph structure. The motivation to use the Gromov-Wasserstein discrepancy is sound, and the chosen relaxations (resistance distance instead of shortest path, orthogonal GW) are necessary to make the certificate tractable. The technical contributions are significant. Since the authors also propose a complementary attack it is possible to experimentally evaluate the tightness of the proposed certificate by looking at the fraction of non-verifiable graphs (neither robust nor non-robust).

The main weakness of the paper is the experimental evaluation. The certificates and attacks are evaluated only fora a GCN model using a single linear convolutional layer followed by average pooling (see Q1). Moreover, most of the experiments are performed with a fixed local budget which might be over-constraining the set of solutions leading to false sense of security w.r.t. tightness (see Q2).

While the final threat model is definitely a move in the right direction it is not as interpretable as simple edge perturbations. It would be helpful if the authors can provide more intuition (see Q3 and Q4).

The authors do not discuss whether there is an accuracy vs. robustness tradeoff (see Q5 and Q6). While they do show the graph classification performance in section 5.1. these are for different models, and not the one they are certifying. Relatedly, I think that sections 5.1. and 5.2 distract the reader from the main message. While they provide interesting insights that should perhaps be relegated to the appendix. The additional space can be used to address some of the questions below.

---

> ### Author Response · Authors · 2022-08-02
> **Rebuttal**
>
> We thank the reviewer for the constructive comments.
>
> **Q1:** Can the certificate be applied to multi-layer GCNs, and if yes, how do the results change?
>
> **A:** It is noteworthy that our relaxation of OGW with resistance distance is orthogonal to the graph classifier.
> When the classifier is changed, one will only need to re-derive the expression of $M^{\*\*}$ for the risk/margin $M$.
> The constraint in Eq 30 and 32, however, remain unchanged, and that part is the contribution of our paper.
> We experimented with single-layer GCN for $M$,
> but orthogonal to our contribution of OGW relaxation,
> one can develop $M^{\*\*}$ for multi-layer GCNs.
> In fact, Appendix D of [7] has discussed it but no experiment or code implementation is given.
> We conjecture that a higher gap will be found in $M^{\*\*}$ for multi-layer GCN,
> and the overall certificate will be less tight.
>
>
> **Q2:** How do the plots in Fig. 3 change for larger values of the local budget?
>
> **A:** Figure 3 is under unlimited local budget.
> If we enforce a finite local budget, the gap between OGW and $\Omega$ will only get narrower.
>
> **Q3, Q4:** Given very large (practically infinite) values of local and global budget and some seed original graph, what are all the reachable graphs that satisfy the OGW constraint? It would be helpful to provide a few examples, starting with small values of $\delta_\Omega$ and gradually increasing it. Next, also enforce the local and global budgets.
>
>
> **A:**
> Due to the content limitation of OpenReview, we placed our new figures in the revised Supplementary Material as Appendix C.
>
>  Without the constraints of the local and global budget, checking all the reachable graphs essentially finds all possible perturbations under the budget $\delta_\Omega$ on $\Omega$, which is (NP) hard.
>  Alternatively, we examined the $\Omega$ value on the real dataset MUTAG by extracting all the graphs with 12 nodes,
>  and then presented their pairwise $\Omega$ distance (first line in each cell) and $\delta_g$ (second line) in Fig 10.
>
>  To better visualize the result,
>  Fig 11 and 12 set the budget $\delta_\Omega$ to $0.5$ and $1$ respectively,
>  and a darker shade represents a higher value of $\Omega$.
>  A cell is marked with two numbers (red for $\Omega$ and black for \#perturbed-edge) computed from a pair of reachable graphs,
>  if its $\Omega$ value falls below the $\delta_\Omega$ budget.
>  In Fig 11,
>  we observe that in the first row, columns 5, 10, 11, 12, and 13 exhibit high values of \#perturbed-edge, but their $\Omega$ value is 0.
>  In these cases, the pair of graphs are isomorphic, although their topology differs a lot.
>  We also see a block of four isomorphic graphs in the bottom-right corner.
>  Comparing Fig 12 with Fig 11,
>  clearly more pairs of graphs become reachable thanks to the increase in $\delta_\Omega$.
>
>
>  Similarly, Fig 13 and 14 set the threshold of $\delta_g$ to $4$ and $8$ respectively,
>  and a darker shade represents a higher value of \#perturbed-edge.
>  A cell is marked with two numbers (red for \#perturbed-edge and black for $\Omega$) computed from a pair of reachable graphs,
>  if its \#perturbed-edge falls below the $\delta_g$ budget.
>
> **Q5, Q6:** What is the accuracy of the certified model in section 5.3. and how does it compare to SOTA (non-certifiable) models? How do vanilla-trained models perform and does robust training decrease the clean accuracy?
>
> **A:** We compared our vanilla and robust one-layer GCN model with two SOTA models,
> namely MemGNN [a] and FactorGCN [b].
> To be consistent with our setting,
> we split the train, validation and testing sets into 30, 20, and 50\% respectively.
> All the other hyperparameters follow the standard setting from the paper.
> The table below reports the average accuracy on the testing set with 5 runs.
>
>  | Dataset | Our vanilla model | Our robust model | MemGNN | FactorGCN |
>  | ------ | :----: | :--: | :------: | :------: |
>  | BZR    | 78.4| 77.5| 84.7   | 82.4|
>  | COX2   | 78.3| 76.9| 79.0   | 81.9      |
>  | MUTAG  | 65.4| 64.8| 77.8   | 82.6|
>  | PTC_MR | 57.6| 51.9| 59.8   | 54.6|
>
> So it can be observed that most times the robust model sacrifices only a slight amount of accuracy compared with our vanilla model (Q6).
> Our vanilla model is indeed inferior to SOTA, though pretty close on COX2 and PTC\_MR.
> Although it will be ideal if we can certify for SOTA models,
> it proves very challenging and we would like to overcome the difficulties step by step.
> It is once more noteworthy that constructing convex relaxations of these SOTA models is orthogonal to our contribution, which is to tightly relax OGW with resistance distance.
> The robust training model is also commonly used in literature, e.g., [5, 7, 36].
>
> [a] Khasahmadi, Amir Hosein, et al. "Memory-Based Graph Networks." International Conference on Learning Representations. 2020.
>
> [b] Yang, Yiding, et al. "Factorizable graph convolutional networks." Advances in Neural Information Processing Systems. 2020.

---

> > ### Comment · Reviewer_hBdY · 2022-08-08
> > **Thanks for the response**
> >
> > Thank you for the comprehensive response. All of my questions have been adequately addressed. I have one follow up question/comment: It would be helpful to show the performance of a baseline that only uses the node features (e.g. the one-hot encoding of the atom type for the MUTAG dataset) while ignoring the graph structure. This model will be perfectly robust w.r.t. any kind of structure perturbation and any threat model. Specifically, since the accuracy of the vanilla/robust model is quite a bit lower compared to MemGNN and FactorGCN it could be that e.g. an MLP on node features only can provide a better (or at least different) trade-off.

---

> > > ### Author Response · Authors · 2022-08-09
> > > **Thank you for proposing the new baseline**
> > >
> > > It is interesting to compare with an MLP trained with node feature only.  We conducted the experiment, and the result is shown in the fifth column of the table below.  Interestingly, we also noticed that our previous experiment on single-layer GCN can enjoy slight improvement in accuracy with more refined hyperparameter tuning.  So we also list the new results of GCN in columns 3 and 4, with the previous GCN accuracy results retained in columns 1 and 2.
> > >
> > > It can be observed that GCN’s accuracy is improved by about 2-3\%, making both the vanilla and robust versions (columns 3 and 4) more accurate than node-feature based MLP.  We redid the attack and certificate for the MUTAG dataset under the new hyperparameter values, and the results are given in Figure 15 of Appendix D in the updated Supplementary Material.  Clearly for both $\delta_\Omega=1$ and $\delta_\Omega=10$, the gap that corresponds to undetermined graphs is significantly **narrower** than that in Figure 5, demonstrating even better tightness of certificate and attack.  This is not surprising because models with higher accuracy tend to have a larger margin, hence easier for the certification of robustness.
> > > Due to time constraint, we will continue to perform similar experiments on the other datasets and update the paper accordingly.
> > >
> > > In practice, hyperparameter tuning for neural networks is known to be challenging, with much ongoing research.  Therefore, a perfectly trained model with perfectly tuned hyperparameter is not necessarily the only model whose robustness is of concern.
> > >
> > >
> > >  | Dataset | Vanilla-GCN (old) | Robust-GCN (old) | Vanilla-GCN (new) | Robust-GCN (new) |  MLP  | MemGNN | FactorGCN |
> > >  | ------- | :---------------: | :--------------: | :---------------: | :--------------: | :---: | :----: | :-------: |
> > >  | BZR     |       78.4        |       77.5       |       81.8        |       80.3       | 79.9  |  84.7  |   82.4    |
> > >  | COX2    |       78.3        |       76.9       |       79.9        |       78.6       | 78.2  |  79.0  |   81.9    |
> > >  | MUTAG   |       65.4        |       64.8       |       69.5        |       67.4       | 65.0  |  77.8  |   82.6    |
> > >  | PTC_MR  |       57.6        |       51.9       |       57.8        |       57.8       | 57.3  |  59.8  |   54.6    |

---

> > > > ### Comment · Reviewer_hBdY · 2022-08-09
> > > > **Thank you**
> > > >
> > > > Thank you for including the additional baseline. Given how close the perfectly-robust MLP is to the (new) Robust-GCN in terms of accuracy, the Robust-GCN might not be the most useful model in practice since the accuracy vs. robustness tradeoff might not be most favorable.
> > > >
> > > > Nonetheless, I think your work and the proposed model is an important first step towards better certified models and therefore I support the acceptance of this paper.

---

### Official Review · Reviewer_yoMg · 2022-07-11

**Rating:** 7
**Confidence:** 3
**Soundness:** 3 good
**Presentation:** 3 good
**Contribution:** 4 excellent

**Summary:**

The papers consider certifiable certificates under the Orthogonal Gromov Wasserstein discrepancy (OGW). This contrasts with other graphs “distances” used in the literature, such as L1 distance between the adjacency matrices. The advantage of OGW is that it considers symmetries/isometries and does not rely on a fixed node ordering. To compute the certificate, the authors use convex relation. More specifically, they propose using the resistance distance as the metric space on the graph (as opposed to the more commonly used shortest path metric) and design a convex relaxation of this computation. Secondly, they give a convex relation of the OGW discrepancy measure. The authors demonstrate the certificate on a single-layer GCN graph classifier.

**Questions:**

I have a few small questions:

- “bona fide distance”, do the authors just mean metric?
- “but the Kantorovich dual no longer exists” (line 36), does this line mean it exists for the Gromov Wasserstein distance but not the discrepancy?
- Why setting Z is equivalent to (20)?
- Why does Z1 = -1?
- Line 105, what is the spectral constraint of \mathcal{E}? Can the authors comment on why them being a spectral constraint is a reason to use just them?

**Limitations:**

The main limitations of the proposed method are that is does not scale that well, and it is not clear how it would perform with models beyond the very simple model they consider.

**Strengths And Weaknesses:**

The paper addresses an interesting and important problem of certifying a graph classifier. The problem is extended beyond the usual global and local budget to consider a budget under the Gromov Wasserstein discrepancy (OGW). To the best of the authors knowledge, as well as my knowledge, they are the first to consider an extension to a budget that considers isometry. I do wonder however, if when combined with local and global budgets, if this can be taken advantage of. For example, if many edges are flipped to give an isometric graph, the OGW will be zero, but the local and global budgets will still be violated.

In general I think the experimental section is well written. The authors do a good job of motivating the use of effective distance, and give a convincing argument for its use by through SVM classifiers and barycenter visualisation. Unfortunately, the authors only experimented with a very simple model. As I understand it the model is a GCN convolutional layer with 64 hidden units, followed by an averaging pooling and presumably a linear layer afterwards, with no activation functions. This can be written as 1/n \mathbf{1} A X W_1 W_2 where  1/n \mathbf{1}  gives the pooling, A X W_1 is the GCN layer and W_2 is the final layer. Since there are no non-linearities, we can tie the weights W_1 W_2 to get an equivalent model and see its equivalent to a GCN with a single hidden unit followed by an averaging of the node outputs. It's not actually clear to me if this certificate could be used on more complex models without further modification and, if so if it would become too loose to be a useful certificate. It would be appreciated if the authors could comment on this or even better demonstrate their method on a slightly more complex model.

The other weakness is the scalability of the certificate. The authors are upfront in the paper that the certificate is O(n^3). As shown in figure 9, the certificate takes a few minutes to compute, even for very small graphs. I wonder if it's feasible to run a certificate like this on something like COLLAB graphs? Would the authors consider extending figure 9 or commenting on the sorts of time required for graphs of sizes 50, 100 or 200 nodes?

have some other minor comments.
- Might help to point out that the coupling is the set of doubly stochastic matrices (line 104).
- Expanding on some of the derivations in an appendix would be useful. For example, I could not follow lines (11) and (12). There was also some parts in Section 3.2 I did not follow, for example why setting Z is equivalent to (20). And why Z1 = -1.
- Might be useful to clarify what “Vec. Attr.” and “Disc. Attr.” mean in table 2.

In general, I think the paper is original and of high quality. Despite the weaknesses I've mentioned, I think it is a good paper demonstrating how to use OGW in a certificate. I think this paper can inspire future research into certificates that consider graph distances/discrepancies beyond those that assume fixed node orderings and that this is important as most GNNs are invariant to node reorderings (i.e. node permutation)

---

> ### Author Response · Authors · 2022-08-02
> **Rebuttal**
>
> We thank the reviewer for the constructive comments.
>
> **Q1:** “bona fide distance”, do the authors just mean metric?
>
> **A:** Yes we mean metric.
>
> **Q2:** “but the Kantorovich dual no longer exists” (line 36), does this line mean it exists for the Gromov Wasserstein distance but not the discrepancy?
>
> **A:** Kantorovich dual is available for Wasserstein distance, but it does not exist for Gromov-Wasserstein distance or discrepancy.
>
> **Q3:** Why setting Z is equivalent to (20)?
>
> **A:** Line 174 writes that the minimum value of $\ell(Z,\Phi)$ is 0, and is attained if and only if $Z = \nabla F(\Phi)$.
> So enforcing $\ell \le 0$ is equivalent to setting $\ell=0$,
> which in turn implies $Z = \nabla F(\tilde{L}+\frac{1}{n} \mathbf{1} \mathbf{1}^\top) = -(\tilde{L}+\frac{1}{n} \mathbf{1} \mathbf{1}^\top)^{-1}$.
>
> **Q4:** Why does Z1 = -1?
>
> **A:** $Z = -(\tilde{L}+\frac{1}{n} \mathbf{1} \mathbf{1}^\top)^{-1}$.
> It is known that $\tilde{L}$ is the Laplacian for a connected graph.
> So it has an eigenvalue 0 with eigenvector $\mathbf{1} / \sqrt{n}$,
> and the remaining eigenvalues are nonzero whose corresponding eigenvectors are orthogonal to $\mathbf{1}$.
> As a result, $Z$ has an eigenvalue $-1$ with eigenvector $\mathbf{1} / \sqrt{n}$.
>
>
> **Q5:** Line 105, what is the spectral constraint of $\mathcal{E}$? Can the authors comment on why them being a spectral constraint is a reason to use just them?
>
> **A:** spectral constraint means it constrains a matrix only through its spectrum (eigenvalues and eigenvectors).
> For example, $\mathcal{O}$ constrains that all eigenvalues are 1,
> and $\mathcal{E}$ constrains that the matrix has an eigenvector $\mathbf{1}/\sqrt{n}$ whose corresponding eigenvalue is 1.
> If we only employ spectral constraints,
> then we can focus on optimizing the spectrum and get simpler optimization problems.
> For example there is a closed-form solution to $Z$ as presented in Appendix A.3.
>
>
> **Q6:** running time on larger graphs.
>
> **A:** We empirically measured the walltime for the graphs from COLLAB.
> The time cost per iteration on COLLAB with graph size of 50, 100 and 200 is approximately 45, 100 and 250 seconds on average, respectively.
>
> **Q7:** It's not actually clear to me if this certificate could be used on more complex models without further modification and, if so if it would become too loose to be a useful certificate. It would be appreciated if the authors could comment on this or even better demonstrate their method on a slightly more complex model.
>
> **A:** It is noteworthy that our relaxation of OGW with resistance distance is orthogonal to the graph classifier.
> When the classifier is changed, one will only need to re-derive the expression of $M^{\*\*}$ for the risk/margin $M$.
> The constraint in Eq 30 and 32, however, remain unchanged, and that part is the contribution of our paper.
> We experimented with single-layer GCN for $M$,
> but orthogonal to our contribution of OGW relaxation,
> one can develop $M^{\*\*}$ for multi-layer GCNs.
> In fact, Appendix D of [7] has discussed it but no experiment or code implementation is given.
> We conjecture that higher gap will be found in $M^{\*\*}$ for multi-layer GCN,
> and the overall certificate will be less tight.

---

> > ### Comment · Reviewer_yoMg · 2022-08-08
> > **Follow up on A5/A7**
> >
> > I thank the authors for their response.
> >
> > **Response to A5**
> > The concept of spectral constraints is clear to me. However, I was hoping for a more technical explanation of how these sets correspond to their respective constraints.
> >
> > My understanding is that $\mathcal{O}$ constrains the eigenvalue to sit on the complex unit circle. Why do the authors say it constrains the eigenvalues to 1?
> >
> > Can the authors explain why $\mathcal{E}$ gives these spectral constraints?
> >
> > **Response to A7**
> > This paper is a good contribution, even if the experiments are for a single-layer GCN. However, as other reviewers have also picked up, it highlights a weakness in this method of certification. Not scaling with depth is inherent to convex relation certification methods and not specific to this method. However, a discussion about the possibility of this extension and its implications (like given in the response) would improve the paper whilst requiring minimal changes.

---

> > > ### Author Response · Authors · 2022-08-08
> > > **Thank you and response**
> > >
> > > We thank the reviewer for the discussion.
> > >
> > > Regarding **A5**: yes, $\mathcal{O}$ constrains the eigenvalue to the complex unit circle.  $P$ is not symmetric in general; even if so, the eigenvalue can be $-1$.  We are sorry for the oversight. This is a spectral constraint, i.e., expressed exclusively in terms of eigenvalues or eigenvectors.  The key benefit of it is that in Eq 11, both the optimal $Q_1$ and $Q_2$ can be found in a closed form where they are constrained to $\mathcal{O}_{n-1}$.  The solution is based on SVD and eigen-decomposition.
> > >
> > > $\mathcal{E}$ stipulates $P \mathbf{1} = P^\top \mathbf{1} = \mathbf{1}$, meaning, by definition, that $\mathbf{1}/\sqrt{n}$ is an eigenvector of both $P$ and $P^\top$ with eigenvalue 1.  These are also constraints on the eigenvalues and eigenvectors, hence spectral constraints.  So it works quite well with $\mathcal{O}$ as shown in the derivation of $OGW$ lower bound in Section 2.2.  It also allows the $Z$ in objective Eq 37-38 to be solved in closed form.
> > >
> > > In contrast, constraints like $P_{ij} \ge 0$ are not spectral.  So optimization (e.g., minimizing a linear function) gets harder when it is combined with spectral constraints, e.g., $P P^\top = I$.  Clearly, spectral constraints do not always render a closed-form solution; an example is Eq 9. But distinguishing spectral constraints from non-spectral ones may potentially provide some clue on the difficulty of optimization.
> > >
> > > Regarding **A7**: this is a great point, thank you.  Our method is limited in the sense that it requires a tight convex lower approximation of the margin under the given classifier.  Although it is orthogonal to our contribution of relaxing the constraint set (threat model), we will include our response in the revised paper and clarify such a limitation, along with a discussion about the possible extensions and implications.

---

> > > > ### Comment · Reviewer_yoMg · 2022-08-09
> > > > **Response**
> > > >
> > > > Thank you for explaining the spectral constraint of $\mathcal{E}$, it's quite simple and an oversight on my part. I thank the authors for providing an insightful discussion on their very interesting paper and answering all my questions and concerns.

---

### Official Review · Reviewer_cA6U · 2022-07-12

**Rating:** 5
**Confidence:** 2
**Soundness:** 2 fair
**Presentation:** 2 fair
**Contribution:** 2 fair

**Summary:**

In this paper, the authors consider the certification and attack of graph convolution network-based classifier.
In this work, the authors proposed an orthogonal Gromov-Wasserstein (OGW )discrepancy to quantify the strength of the attack on graph topology.
The OGW achieves a convex approximation of the original GW distance, which can be computed efficiently.
Moreover, OGW yields a tight outer convex approximation for resistance distance on graph nodes.
Experimental results demonstrate that the OGW-based resistance distance can be used to achieve kernel-based graph classification and works better than the shortest path-based method.
Additionally, the authors also verified the rationality of using OGW-based threats to attack and certification of GCN classifiers.

**Questions:**

(1) Line 109, Eq.(7), what is the difference between E and \hat{E}?

(2) Line 134, Eq.(14), why do we need the notation "f^{**}"?

(3) Line 142, what is the difference between D and \mathcal{D}?

(4) Line 221, what is the definition of Omega^{*}? What is the difference between M^* and M^**?

---- after rebuttal ----
The authors explained the logic behind their writing and revised the submission based on some of my comments.
Although I am not an expert in this field, I am willing to change my score to 5.



**Limitations:**

Although I am not an expert in this field, I believe that this paper should be rewritten completely. Its current status is not reading-friendly.

**Strengths And Weaknesses:**

Weaknesses:

(1) I am not an expert on model certification and attack, but I think the organization of the paper is questionable. I don't think introducing OGW in section 2 is a good idea. The authors should introduce the target problem or the task before introducing the technique part, such that the readers can obtain a big picture about what the authors did and what key challenges they solved.

(2) There are many notations used without definitions, which makes the paper hard to follow. (See the questions below)

---

> ### Author Response · Authors · 2022-08-02
> **Rebuttal**
>
> We thank the reviewer for the constructive comments.
>
> **Q1:**
> Line 109, Eq.(7), what is the difference between $E$ and $\hat{E}$?
>
> **A:**
> In line 109, we denoted the $\hat{X}$ for **any** matrix $X$ .
> $\hat{E}$ is the projection of $E\in R^{n \times n}$ into a lower dimensional space $\hat{E} \in R^{(n-1) \times (n-1)}$.
>
> **Q2:**
> Line 134, Eq.(14), why do we need the notation $f^{\*\*}$?
>
> **A:**
> $f^{\*\*}$ is the Fenchel biconjugate of $f$, which is the conjugate of $f^\*$.
> $f^*$ (Fenchel conjugate) is introduced in Eq 10.
> This is analogous to the notation that $f'$ is the derivative of $f$,
> while $f''$ is the derivative of $f'$.
>
> **Q3:**
> Line 142, what is the difference between $D$ and $\mathcal{D}$?
>
> **A:**
> $\mathcal{D}$ refers to a graph,
> while $D$ is an $n$-by-$n$ matrix encoding the distance measure between all pairs of nodes in $\mathcal{D}$.
> This notation was introduced around Eq 1 (line 88, 89, 93, 94).
>
> **Q4:**
> Line 221, what is the definition of $\Omega^*$?
> What is the difference between $M^*$ and $M^{**}$?
>
> **A:**
> $\Omega^*$ and $M^*$ are the Fenchel conjugate of $\Omega$ and $M$, respectively.
> Fenchel conjugate is introduced in Eq 10.
> $M^{**}$ is the Fenchel biconjugate of $M$ as mentioned in Q2.
>
>
> **Q5:**
> I am not an expert on model certification and attack, but I think the organization of the paper is questionable. I don't think introducing OGW in section 2 is a good idea. The authors should introduce the target problem or the task before introducing the technique part, such that the readers can obtain a big picture about what the authors did and what key challenges they solved.
>
> **A:**  The "big picture" of the threat model, as given in Eq 17, relies on the definition of OGW.
> So we can at most move the lower bound and upper bound paragraphs from Section 2 to later.
> However, it is surely more natural to introduce these bounds right after OGW is introduced,
> at the cost of slightly delaying the presentation of the threat model Eq 17.
> We think both are reasonable approaches,
> with our current presentation placing our relaxation of OGW on a more general footing,
> instead of specifically designed for robustness certificates.
> After all, the "big picture" has also been furnished in the introduction.

---

> > ### Comment · Reviewer_cA6U · 2022-08-07
> > **Reply to the authors, AC, and SAC**
> >
> > (1) Thanks for explaining the notations of the submission, and I think these explanations are necessary for the revised paper.
> > Additionally, I believe that the notations and the symbols in this submission can be much simplified, which will definitely make the submission more reading-friendly.
> >
> > (2) I still think the organization of the paper is questionable and can be improved. In my opinion, the threat model is a general concept, and its OGW-based implementation in Eq. 17 is an instantiation of the concept. I suggest the authors consider the following logic flow: (a) first, introduce what model certification and model threatening are in graph classification scenarios; (b) second, show a general form of Eq. 17, e.g., we need to find a metric $d$ for graphs, such that $d(C_{A+X}, D)\leq \delta_{\Omega}$; (c) GW provides a natural choice of $d$, and the main contribution is proposing a variant of GW, called OGW. (d) Finally, introducing the technical part of OGW, e.g., its formulation, computation, and theoretical guarantees.
> >
> > Again, I am not an expert in this field, so I set a low confidence score for my review. Let me know if my understanding is incorrect. It would be nice if AC or SAC join in the discussion and evaluation.
> >
> > To AC/SAC: feel free to ignore my review if it is incorrect:)

---

> > > ### Author Response · Authors · 2022-08-08
> > > **Paper revised**
> > >
> > > We thank the reviewer for the detailed and constructive comments.  We have revised the paper following your suggestions.  In the new Section 2, now line 82 to 109 cover exactly your recommended points (a) to (c), with GW rigorously defined in Section 2.1.  After that, OGW is introduced in Section 2.2 (point d), and the resulting attack model is immediately introduced in Eq 7.
> > >
> > > Incidentally, we wish to clarify that OGW was proposed by [24]. Our contribution is not to propose it but to identify its advantage in robustness certification under isomorphic graph threats. This was stated in line 100 of our original submission (dated 19 May).
> > >
> > > We will appreciate it if any concrete suggestions can be offered regarding “I believe that the notations and the symbols in this submission can be much simplified”.  The techniques in this paper are quite involved, and we have made every effort simplifying the notation and presentation.

---

> > > > ### Comment · Reviewer_cA6U · 2022-08-08
> > > > **Reply to the authors after reading the revised paper**
> > > >
> > > > Thanks for your quick update. For simplifying notations, I have some quick suggestions:
> > > >
> > > > (1) Represent matrices via a bolden and uppercase format; represent their elements via a regular, lowercase format with subscripts; represent sets via the "mathcal" type; use "\hat" and "\tilde" more carefully... and moreover, keep the notations consistent in the whole paper.
> > > >
> > > > (2) Some notations are useless/ignorable because they either appear only once or are insignificant, e.g., the $\mathcal{N}$ in Eq.(4), the graph notations $\mathcal{C}$ and $\mathcal{D}$ in Line 112, the $\Pi$ in Eq.(5), the $\Gamma(A+X_{\lambda})$ at Algorithm 1 and 2, etc.
> > > >
> > > > (3) Eqs.(12-15), pls use $\mathcal{O}_n$ consistently.
> > > >
> > > > (4) In line 186, $\pi$ actually is equivalent to the notation $s_{\tilde{A}}$, which has already been defined.
> > > >
> > > > Anyway, I believe that more improvements can be made.
> > > >
> > > > Additionally, as the authors mentioned,  the main contribution of this submission is identifying the advantage of OGW in graph classifier certification. I am not sure whether it is a significant methodological contribution.

---

> > > > > ### Author Response · Authors · 2022-08-08
> > > > > **Thank you and response**
> > > > >
> > > > > We are very grateful for the prompt response and for the detailed suggestions.
> > > > >
> > > > > We wish to first point out a key misunderstanding in your comment about our main contribution.  We wrote that it is our contribution to identify the advantage of OGW in graph classifier certification.  However, far from our “main contribution”, it is only a minor part of it.  In fact, our major contributions have been highlighted in line 45 to 57, which are reiterated here:
> > > > >
> > > > > 1. designing a convex relaxation for resistance distance in Section 3 based on the matching loss;
> > > > > 2. designing a convex lower bound for OGW in Section 2.3;
> > > > > 3. showing that the two lower bounds can be efficiently computed and provide a jointly tight relaxation for robustness certification.
> > > > >
> > > > > We have revised the paper by adding the subscript $n$ to $\mathcal{O}$ and rephrasing the graph volume sentence to eliminate the symbol $\pi$.   We are grateful for these points.  With all due respect, we disagree with the rest of the comments for the following reasons:
> > > > >
> > > > > 1. We have consistently represented matrices by uppercase letters (English or Greek).  This is the standard practice in optimization (see the latest issue of the Mathematical Programming journal or the “Convex Optimization” book by Boyd and Vandenberghe) and is also commonly used in machine learning [e.g., 1, 7, 8, 11, 12, 13, 16, 22, 23, etc]. Although bolden with uppercase is also commonly used in machine learning, we find it a bit too obtrusive, and uppercase is sufficient to signify it is a matrix.  When its elements are indexed, we simply add subscripts and the matrix symbol itself needs no change (no bold to drop).  Most literature in convex optimization (heavily overlapping with the machine learning community) do not switch to lowercase when referring to a matrix’s elements. For example, the “Convex Optimization” book by Boyd and Vandenberghe.
> > > > >
> > > > > 2. We have generally used “mathcal” to represent sets.  One exception is $\Pi$, which is the set of all permutation matrices.  This is such a textbook-level standard notation that we really don’t want to deviate from.
> > > > >
> > > > > 3. We believe "\hat" and "\tilde" have been used consistently and carefully.  In fact, the \hat notation comes from [24], which is in turn from [c] to denote projection to a subspace.  The \tilde notation neatly indicates that it refers to the new perturbed graph.
> > > > >
> > > > > 4. $\mathcal{N}$ is the standard notation for the non-negative orthant, used extensively in optimization such as [c, d].  It is cleaner than $R_+^{n \times n}$ and resonates well with those who are familiar with this standard notation.
> > > > >
> > > > > 5. $\mathcal{C}$ and $\mathcal{D}$ are also not redundant, because the graphs warrant a symbol, especially when there are two graphs to differentiate.  The standard way in graph theory to introduce a graph is by a tuple of node set and edge set.  We have simplified them by just using natural numbers and the adjacency matrix.  Later on, we also endeavored to simplify notation by writing $GW(C, D)$ instead of $GW(\mathcal{C}, \mathcal{D})$.  But this shouldn’t make $\mathcal{C}$ and $\mathcal{D}$ redundant, because in the text, directly calling a graph by its distance matrix $C$ is still awkward.
> > > > >
> > > > > 6. $\Pi$ helps to much simplify the presentation in line 127, 128, and Eq 5.  In addition, as mentioned above, bringing up such a textbook-level notation would probably not add much burden to comprehension.
> > > > >
> > > > > 7. $\Upsilon (A+X)$ is introduced because it significantly simplifies the long expression $OGW_{ub}(C_{A+X}, C_A)$.  In addition, Algorithm 1 introduces $A_\lambda$, which would lead to a subscript of a subscript if it were not for such a shorthand.  Since $\Upsilon(A+X)$ is used in line 217 and five times in Algorithm 1 and 2, we believe this shorthand is worthwhile.
> > > > >
> > > > > To summarize, all notations have pros and cons, and sometimes we have to reconcile between the standard notation and what’s consistent within a paper.  Sometimes different fields have different standards of notation, and a paper drawing upon multiple disciplines also has to make reconciliations.  In such a technically involved paper as ours, the only catch spotted so far is the missing subscript $n$ in $\mathcal{O}$, although it obviously did not impede understanding.  We thank the reviewer for engaging in the discussion and making constructive and detailed comments.  We do agree that “It would be nice if AC or SAC [or co-reviewers] join in the discussion and evaluation [on presentation]”.
> > > > >
> > > > >
> > > > > [c] S. W. Hadley, F. Rendl, and H. Wolkowicz. A new lower bound via projection for the quadratic assignment problem. Mathematics of Operations Research, 17(3):727–739, 1992.
> > > > >
> > > > > [d] K. M. Anstreicher and N. W. Brixius. A new bound for the quadratic assignment problem based on convex quadratic programming. Mathematical Programming, 89: 341–357, 2001.

---

> > > > > > ### Comment · Reviewer_cA6U · 2022-08-09
> > > > > > **Thank you for your detailed explanation**
> > > > > >
> > > > > > Thanks again for your quick update and detailed explanation.
> > > > > > Although I still have some concerns about notations and the paper's organization, I appreciate your efforts and understand the logic behind your writing after this rebuttal-discussion phase.
> > > > > > I am willing to raise my evaluation score from 3 to 5 but keep my confidence score unchanged.

---

### Meta-Review · Area_Chair_hzFB · 2022-08-30

**Recommendation:** Accept
**Confidence:** Certain

**Metareview:**

This paper proposes a robustness certificate for graph classifiers under orthogonal Gromov-Wasserstein (OGW) threat models. OGW considers symmetries/isometries and does not rely on a fixed node ordering. The computation of the certificate is based on convex relations. The certificate is demonstrated on a single-layer GCN graph classifier. The paper addresses the interesting and important problem of certifying a graph classifier. The reviewers found the paper original and of high quality. During the rebuttal period, the authors provided an insightful discussion on their work and addressed most of the questions and concerns raised by the reviewers.

**Award:**

No

---

### Decision · Program_Chairs · 2022-09-14

Accept